



# Improvement of the soil drainage simulation based on observations from lysimeters

Antoine Sobaga[1,2], Bertrand Decharme[2], Florence Habets[1], Christine Delire[2], Noële Enjelvin[3], Paul-Olivier Redon[4], Pierre Faure-Catteloin[5], and Patrick Le Moigne[2]

[1]Laboratoire de Géologie - CNRS UMR 8538 - École Normale Supérieure - PSL University, IPSL, Paris, France
[2]Centre National de Recherches Météorologiques, Université de Toulouse, Météo-France, CNRS UMR 3589, Toulouse, France
[3]Laboratoire Sols et Environnement-GISFI, Université de Lorraine (UMR 1120), Vandœuvre-lès-Nancy, France
[4]Andra, Direction RD, Centre de Meuse/Haute-Marne, 55290 Bure, France
[5]Université de Lorraine, CNRS, LIEC, F-54000 Nancy, France

**Correspondence:** Antoine Sobaga (sobaga@geologie.ens.fr)

**Abstract.** Soil drainage is the main source of groundwater recharge and river flow. It is therefore a key process for water resource management. In this study, we evaluate the soil drainage simulated by the Interaction-Soil-Biosphere-Atmosphere (ISBA) land surface model currently used for hydrological applications from the watershed scale to the global scale. This validation is done using seven lysimeters from two long term experiment sites measuring hourly water dynamics between 2009 and 2019 in northeastern France. These 2-meter deep lysimeters are filled with different soil types and are either maintained bare soil or covered with vegetation. The commonly used closed-form equations describing soil-water retention and conductivity curves from Brooks and Corey (1966) and van Genuchten (1980) are tested. The results indicate a good performance by the different experiments in terms of soil volumetric water content and water mass. The drained flow at the bottom of the lysimeter is well modeled using Brooks and Corey (1966) while some weaknesses appears with van Genuchten (1980) due to the complexity of its hydraulic conductivity function. Combining the soil-water curve of van Genuchten (1980) with the hydraulic conductivity function of Brooks and Corey (1966) allow to solve this problem and even to improve the simulation of the drainage dynamic, especially for intense drainage events. The study highlights the importance of the vertical heterogeneity of the soil hydrodynamic parameters to correctly simulate the drainage dynamic, as well as the primary influence of the $n$ and $b$ parameters which characterize the shape of the soil-water retention curve.

## 1 Introduction

Drainage is the portion of precipitation that flows through the first meters of soil. As it has escaped evapotranspiration, it is the main diffuse source of groundwater recharge (Moeck et al., 2020), and is of crucial importance for estimating the evolution of the aquifer (Bredehoeft, 2002; Döll and Fiedler, 2008). Even where there is no aquifer, this water flux can contribute to the baseflow. Despite their importance, direct observations of drainage are still rare compared to direct observations of river flow or groundwater level.




Indirect methods based on analysis of baseflow (Meyboom, 1961), or variation of piezometric level for instance using a water table fluctuation method (Healy and Cook, 2002)) can be applied to estimate groundwater recharge. However, these methods cannot separate the different components of the recharge, such as the exchange between surface water and groundwa-
ter (Brunner et al., 2017; Keshavarzi et al., 2016), exchanges between several aquifer layers basins (Tavakoly et al., 2019), and drainage. Direct measurement at the local scale and high frequency can be made in situ using lysimeter columns, which isolate a small volume of soil from lateral flow and collect drainage. Most of the time, the lysimeters are disconnected from the deep soil and avoid the capillary rise from deeper soil that can have an important influence locally (Vergnes et al., 2014; Maxwell and Condon, 2016). From such observations, drainage is known to have large variations in space, due to changes in soil texture
and structure (Vereecken et al., 2019; Moeck et al., 2020).

Due to the limited number of observations and the difficulty to indirectly quantify it, the estimation of the groundwater recharge is one of the twenty-three unsolved problems in hydrology (Blöschl et al., 2019). The simulation of recharge in hydrological models can vary significantly: from simplified reservoir approaches to physically based models. The widely used
reservoir approach (Alcamo et al., 2003; Harbaugh, 2005) has the advantage of limiting the number of parameters to be calibrated and reducing the numerical cost of simulations. Physically based approaches are more complex and are commonly used in Land Surface Models (LSMs) in which vertical water and energy balances between the land surface and the atmosphere can be calculated (De Rosnay et al., 2003; Blyth et al., 2010; Boone et al., 2000).

Today, LSMs are widely used in hydrological applications in order to study the regional and global water cycle, predict streamflow, and inform water resource management (Haddeland et al., 2011; Vincendon et al., 2016; Schellekens et al., 2017; Gelati et al., 2018; Le Moigne et al., 2020; Vergnes et al., 2020; Muñoz Sabater et al., 2021; Rummler et al., 2022). However, LSMs may have difficulty estimating the dynamics of groundwater recharge, particularly during intense precipitation events. Vereecken et al. (2019) suggested a number of directions for improvement: introduce more physical processes such as preferen-
tial flow along the roots and macropores, improve the representation of soil/vegetation and of soil parameters, and improve the spatio-temporal distribution of precipitation. In LSMs, the Richards equation (Richards, 1931) is used to describe the flow of water in the porous soil due to the actions of capillarity and gravity. Mainly used in the hydrology community, this equation has been widely criticized, in particular for its non-linear form leading to unstable numerical behavior, as well as an overestimation of the effect of capillarity (Nimmo, 2010; Farthing and Ogden, 2017).


In the Richards equation, two closed-form equations are often used to represent the variations of volumetric water content with the matric potential and the hydraulic conductivity in the unsaturated zone: Brooks and Corey (1966) and van Genuchten (1980), hereafter BC66 and VG80 respectively. Historically, the closed-form equations from BC66 are mostly used by the atmospheric community in theirs LSMs, as opposed to the ones from VG80, which are mainly used by hydrologists. BC66
proposed simple analytical power functions of soil-water retention and conductivity based on North American soil observations. These relationships are simple to parameterize and very stable numerically. However, they do not include an inflection





point close to saturation, and are thus not derivable, which leads to problems at the connection with the saturated zone. The consequence is a deviation near saturation of volumetric water content. VG80 proposed an improvement of the BC66 relationships close to water saturation, using more complex analytical forms based on European soil observations. However, the VG80 relationships are less stable than BC66 for coarse-textured soil, mainly because of the complexity of the hydraulic conductivity function (Vogel et al., 2000).

Some studies (Braud et al., 1995; Valiantzas, 2011) have proposed to combine the volumetric water and the matrix potential of the VG80 relationship with the hydraulic conductivity of BC66. In theory, such combination should improve the simulation when the soil water is close to saturation, while preserving the simplicity and numerical stability of the BC66 relationships. Many methods are developed to determine appropriate parameters from one relationship to another (van Genuchten, 1980; Lenhard et al., 1989).

The goal of this study is to use a state-to-the-art LSM to evaluate the benefits of BC66 and VG80 relationships in reproducing soil water mass, volumetric water content and drainage flux observed in seven lysimeters during more than five years. The second goal is to test the approach of Braud et al. (1995) and Valiantzas (2011) that combines the soil water retention of VG80 and the hydraulic conductivity of BC66 to simulate the soil hydrology of these lysimeters. We derive their hydrodynamic parameters directly from observation and compare them with several pedotransfer functions (hereafter PTF) commonly used by LSMs. The LSM used is the multi-layer diffusion version of the Interaction-Soil-Biosphere-Atmosphere (ISBA) model that solves directly the mixed form of the Richards equation (Boone et al., 2000; Decharme et al., 2011).

The experimental protocol, including descriptions of both the lysimeters data and the ISBA model, is given in section 2. Soil parameters estimation based on lysimeters observations is described in section 3, while the main results of each experiment are presented in section 4. Finally, a general discussion and the main conclusions are given section 5.

## 2 Experimental Protocol

### 2.1 Data

Seven lysimeters of two French experimental sites located in north-eastern France are used in this study: 4 lysimeters from the GISFI station at Homécourt (49°21'N, 5°99'E) with data record from 2009 to 2016 (hereafter G1, G2, G3, G4) and 3 lysimeters from OPE experimental station close to Osne-le-Val (48°5092'N, 5°2119E) with data record from 2014 to 2019 (hereafter O1, O2, O3). These two sites are separated by a distance of 97 km. The soils are classified according to the World reference base for soil resources WRB (Hazelton and Murphy, 2016).

The GISFI experimental station focuses on the understanding of pollution evolution and the development of decontamination solutions (Lemaire et al., 2019; Huot et al., 2015; Rees et al., 2020; Ouvrard et al., 2011). It participates in the collaborative study TEMPOL (Observation sur le long TErMe de sols POLlués) with the German observatory infrastructure TERENO (Ter-



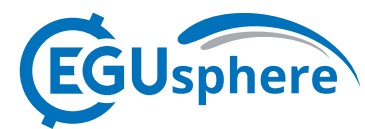

restrial Environmental Observatories, Zacharias et al. (2011)) in order to study in situ the transfer of pollutants (Leyval et al., 2018). Three of the GISFI lysimeters (G1, G2 and G3) contain rebuilt soil of the Spolic Toxic Technosol which was sampled in Neuves-Maisons, an industrial wasteland of a coking plant with contamination (PAH, hydrocarbons, Monserie et al. (2009); Ouvrard et al. (2011); Sterckeman et al. (2000)). These three lysimeters were filled in september 2007 and soil material was gradually and manually compacted every 0.3 m to reach a given bulk density. Lysimeters G1 and G2 were maintained

bared soil, while lysimeter G3 was covered by vegetation (Alfalfa). Lysimeter G4 was filled by a monolith of Cambisol from Noyelles-Godault and covered by grass (Table 1).

The main objective of the OPE site is to describe the environment before and after the construction of the surface facilities of a deep repository for radioactive waste and to follow its evolution. This site is part of the OZCAR (Observatoires de la Zone Critique: Application et Recherche) French research infrastructure dedicated to the observation and study of the critical zone

(Gaillardet et al., 2018; Braud et al., 2020). The lysimeters were filled by monoliths taken closed to the OPE site: lysimeters O1 and O3 were filled with Hypereutric Cambisol with different layers of limestone more or less cracked and lysimeter O2 contains a Cambisol. These three lysimeters had sparse vegetation, composed of C3-type grass (Table 1). Monolith-filled lysimeters preserve original soil horizons and are therefore better representative of watershed conditions.

All the lysimeters are weighable cylinders of 2 m depth and 1 m² area. They are equipped with suction and temperature probes

as well as time-domain reflectometry (TDR) probes to measure the water content, at 0.50, 1.0 and 1.5 m depth, with additional measurement at 0.2 m for the OPE lysimeters. The weight is measured continuously with a 0.1 kg precision giving an idea of the total soil water volume in the column, and tipping counter measures the drainage at the bottom. Data are continuously monitored on an hourly basis using a data logger. A summary of the available data is presented Table 2.

The sets of atmospheric forcing variables (wind speed, precipitation rate, short-wave incident radiation, air temperature, air humidity, atmospheric pressure) are observed in situ at an hourly time step by two local meteorological stations, one at OPE and one at GISFI. Long-wave radiations are derived from the equation of Prata (1996). Atmospheric data gaps are filled by regressions on available data using two neighboring meteorologic stations. The gaps represent up to $15\%$ of the observations for the GISFI site (Table 2). Annual precipitation is $20\%$ higher at OPE compared to GISFI experimental station ($876\,\mathrm{mm.year^{-1}}$,

and $727\,\mathrm{mm.year^{-1}}$, respectively, Table 2). The atmospheric forcing is assumed to be identical for all the lysimeters of each site.

## 2.2 ISBA Model

The ISBA LSM was originally scheduled for use in numerical weather prediction and climate models. These last decades, ISBA has evolved from a simple bucket force-restore model Noilhan and Planton (1989) to a more explicit multi-layer dif-

fusion scheme (Boone et al., 2000; Decharme et al., 2011). ISBA is currently used in hydrological applications, especially to estimate groundwater recharge when it is associated with hydrologic models at both the regional (Le Moigne et al., 2020; Tavakoly et al., 2019; Vergnes et al., 2020) and the global scales (Vergnes and Decharme, 2012; Decharme et al., 2019).



The surface temperature evolves with the heat storage in the soil-vegetation composite and the thermal gradient between sur-
face and the other layers (Boone et al., 2000). At depth, the heat transfer is described by the use of the one-dimensional Fourier
law. The soil heat capacity is the sum of the water heat capacity and the heat capacity of the soil. The soil thermal conductivity
is a function of volumetric water content and porosity. Freezing due to water phase changes in the soil can be also computed
by taking into account the effect of the sublimation of frost and the isolation of vegetation at the surface (Decharme et al., 2016).

ISBA includes an interactive vegetation scheme, activated in this study, that is suitable for 19 types of vegetation. The
scheme represents plant photosynthesis and respiration, plant growth and decay. The simulated stomatal conductance depends
on photosynthesis and controls transpiration. The vegetation canopy is characterized by the leaf area index (LAI), which results
directly from the carbon balance computation of the leaves. The simulated LAI varies during the year according to photosyn-
thetic activity, respiration and decay. In spring for instance, when photosynthetic activity is high due to high solar radiation
and sufficient volumetric water content, LAI increases. LAI affects transpiration and the evaporation of canopy intercepted
rain. Photosynthesis and respiration are parametrized according to the semi-empirical model of Goudriaan et al. (1985) and
implemented by Jacobs et al. (1996) and Calvet et al. (1998). Plant growth and decay are based on Calvet and Soussana (2001),
Gibelin et al. (2006) and Joetzjer et al. (2015). A complete description of the carbon cycle in vegetation can be found in Delire
et al. (2020).

The ISBA soil hydrology (Boone et al., 2000; Decharme et al., 2011) uses the following mixed form of the Richards (1931)
equation to describe the water mass transfer within the soil via Darcy's law :

$$\frac{\partial \omega(z)}{\partial t} = \frac{\partial}{\partial z}\left[(k(z) + v(z))\frac{\partial \psi(z)}{\partial z} + k(z)\right] + S(z) \tag{1}$$

where $\omega$ (m$^3$. m$^{-3}$) is the volumetric water content at each depth $z$ (m), $\psi$ (m) the water pressure head , $k$ (m.s$^{-1}$) the soil
hydraulic conductivity, and $v$ (m.s$^{-1}$) the isothermal vapor conductivity computed via a function of soil texture, volumetric
water content, and temperature (Braud et al., 1995). $S$ (kg.m$^{-2}$. s$^{-1}$) is the soil-water source sink term especially related
to water withdrawn by evapotranspiration. This soil hydrology is solved numerically using a Crank-Nicholson implicit time
scheme where the flux term is linearized via a one-order Taylor series expansion.

## 2.3 Experiments

The 3 main experiments performed use the same ISBA configuration except for the closed-form equations between volumetric
water content, soil matric potential, and hydraulic conductivity, as follows :





- The first experiment, noted BC66, uses the standard version of ISBA, i.e. the following closed-form equations of Brooks and Corey (1966) :

$$\psi(\omega) = psi_{sat}\left(\frac{\omega}{\omega_{sat}}\right)^{-b} \tag{2}$$

$$k(\psi) = k_{sat}\left(\frac{\psi}{\psi_{sat}}\right)^{-\frac{2b+3}{b}} \tag{3}$$

where $b$ represents the dimensionless shape parameter of the soil-water retention curve, while $\psi_{sat}$ (m) and $k_{sat}$ (m.s$^{-1}$) are the soil matrix potential and hydraulic conductivity at saturation, respectively.

- The second experiment, noted VG80, uses the more complex closed-form equations from van Genuchten (1980):

$$\psi(\omega) = \frac{1}{\alpha}\left[\left(\frac{\omega - \omega_r}{\omega_{sat} - \omega_r}\right)^{-n/(n-1)} - 1\right]^{1/n} \tag{4}$$

$$k(\psi) = k_{sat} \cdot \frac{\left((1+|\alpha\psi|^n)^{1-1/n} - |\alpha\psi|^{n-1}\right)^2}{(1+|\alpha\psi|^n)^{(1-1/n)(l+2)}} \tag{5}$$

where $\omega_r$ (m$^3$. m$^{-3}$) is the residual volumetric water content assumed equal to 0 in this study, $\alpha$ (m$^{-1}$) the inflection point where the slope of the soil-water retention curve (d$\omega$/d$\psi$) reaches its maximum value, $n$ a dimensionless coefficient that characterizes the shape of the retention curve, and $l$ the Mualem (1976) dimensionless parameter that determines the shape of the hydraulic conductivity curve.

- The third experiment, noted BCVG, uses the combination of the soil water retention curve from VG80 (Eq. (4)) with the soil hydraulic conductivity from BC66 (Eq. (3)).

Finally, additional sensitivity experiments were made first to compare heterogeneous versus homogeneous soil profiles and secondly to assess performances of soil parameter values derived from usual PTF (Clapp and Hornberger, 1978; Cosby et al., 1984; Carsel and Parrish, 1988; Wosten and van Genuchten, 1988; Vereecken et al., 1989; Weynants et al., 2009) (section 5).

## 3 Estimation of parameters

### 3.1 Soil parameters

The rich data sets collected by the lysimeters allow the derivation of the soil hydrodynamic parameters, such as in previous studies (Brooks and Corey, 1966; van Genuchten, 1980; Carsel and Parrish, 1988). For instance, Fig. 1 plots the volumetric water content, $\omega$, as a function of the logarithm of the absolute value of the soil matrix potential, $\psi$, at each depth for lysimeters G2 and O1, showing a better fit for VG80 than BC66, and heterogeneity with depth. $\omega_{sat}$, $\psi_{sat}$, $\alpha$, $n$, $b$, and $k_{sat}$ can be derived from this observed $\psi$-$\omega$ relationship at each depth of each soil column.





$\omega_{sat}$ is determined by the 99th percentile of the observed soil volumetric water content, $b$ by the slope of the observed $\psi$-$\omega$
relationship, and $\psi_{sat}$ when the observed volumetric water content reaches the saturation, i.e. when d$\omega$/d$\psi$ reaches the maximum value (Clapp and Hornberger, 1978). By assuming the unit-gradient assumption, the drainage at 2 m can be considered equal to the hydraulic conductivity. $k_{sat}$ is then assumed equal to the observed drainage when the soil water content is saturated. This method is compared with another estimate based on the 99.9 percentile of the hourly drainage, and the two methods agree (mean relative deviation < 20 %).


$n$ and $\alpha$ are derived from $b$ and $\psi_{sat}$ using the following simple relationships defined by (van Genuchten, 1980) :

$$n = 1 + \frac{1}{b} \tag{6}$$

$$\alpha = -\frac{1}{\psi_{sat}} \tag{7}$$

Several authors proposed more complex relationships allowing preservation of the capillary effects (Lenhard et al., 1989;
Morel-Seytoux et al., 1996; Sommer and Stöckle, 2010). Ma et al. (1999) compared three methods, and showed that the van Genuchten (1980) and Lenhard et al. (1989) methods gave better results. Note that particular attention is given to the values of $n$ for VG80 because Vogel et al. (2000) determined a limit of $n < 1.3$ below which Eq.(5) is numerically unstable. In this study, we found that our simulations are numerically stable when $n \geq 1.09$, so the limit of $n$ is fixed at $n = 1.09$.

The $l$ parameter in Eq. (5) from VG80 is estimated with a simple calibration via ISBA sensitivity experiments with $l$ ranging from -5 to 5. Better scores are obtained for $l$ equal to 0.5, a classical value, for the OPE experimental station. For the GISFI experimental station, better scores are obtained for $l$ equal -5, which remains consistent with the literature (Wosten and van Genuchten, 1988; Wösten et al., 2001; Schaap et al., 2001).

The derived parameters are presented in Fig. 2. Parameters can vary greatly with depth as well as between lystimeters. The largest differences are found for $b$ (and thus $n$). For all lysimeters except G1 and G4, $b$ increases with depth (and conversely for $n$). For the OPE lysimeters, $b$ starts from 8 at 0.2 m deep, doubles first at 0.5 m and then at 1 m depth, and reaches 50 at 1.5 m. These variations are less pronounced for the GISFI's lysimeters, with only a factor of two at depth for G2. Absolute value of $\psi_{sat}$ decreases with depth (and conversely for $\alpha$) especially for the OPE lysimeters. Comparing our parameter estimates
to PTFs usually used in LSMs (Clapp and Hornberger, 1978; Cosby et al., 1984; Carsel and Parrish, 1988; Wosten and van Genuchten, 1988; Vereecken et al., 1989; Weynants et al., 2009), we find that $b$ (and thus $n$) from in situ estimates are quite different from those determined by the PTFs (boxplots). While $b$ ranges from 5 to 12 with the usual PTFs, our estimates range from 8 to 50. $\psi_{sat}$ (and thus $\alpha$) estimates are slightly different from those determined by usual PTFs. For other parameters, our estimates and the usual PTFs give similar values.



## 3.2 Vegetation parameters

For the lysimeters covered by vegetation (O1, O2, O3, G3 and G4), two additional soil parameters must be determined. The field capacity $\omega_{fc}$ and the wilting point $\omega_{wilt}$ are computed via matrix pressure at -0.33 bar and -15 bar, respectively. The root depths have also been determined. Although rooting depth was not measured at the sites, it is possible to derive a deepening of the roots in the soil with the observations of the volumetric water content profile: if the volumetric water content presents a slow decrease in summer at a given depth, it is considered that the roots have not yet reached this depth. The root depth is thus fixed at 2 m for lysimeters G3 and O2, and varies for lysimeters G4, O1 and O3. From 2009 to 2013, root depth of lysimeter G4 reached 1.5 m; but from June 2013, seeding and harvesting were carried out every year, and the roots are limited to 0.4 m depth. In lysimeters O1 and O3, the root depth reached 0.8 m from 2014 to 2018, and 2 m thereafter.

As there are no measurements of LAI for these lysimeters, the simulation of the LAI can only be compared to the literature. As expected, the simulated LAI is minimal between December and April and maximal in August (not shown). For soils with grass (G4, O1, O2, and O3), the maximum LAI varies between 3 and 5, and the mean annual LAI varies between 1.3 and 2. These values are similar to those found in other studies (Calvet, 2000; Darvishzadeh et al., 2008). LAI for alfalfa cover (G3) is larger ($LAImax = 7.5$, $LAImean = 4.8$) which is expected for such well-developed vegetation (Wolf et al., 1976; Wafa et al., 2018).

## 4 Results

Here, we present the main results for each experiment described in section 2.3 in terms of simulated soil water and drainage dynamics, water budget and intense drainage events. Note that the simulated soil temperatures have also been studied and analysed. All experiments have shown good skill scores ($R^2 > 0.9$) underlying the ability of the ISBA LSM in reproducing observed soil temperatures. Because there are no significant differences between the three experiments compared to the observations, these results are not presented in this study.

### 4.1 Soil water dynamics

The dataset allows assessing the evolution of the total soil water mass derived from the mass of the lysimeter, of the soil volumetric water content at several depth, and of the drainage flux at the bottom of the soil. In the following analysis, periods when the meteorological forcing is reconstructed or when data is of poor quality (Table 2) are not taken into account in the scores computation. These periods are short, with the exception of that of drainage on lysimeter G3 (23 % of the duration).

### 4.1.1 Water Mass

The total water mass in the lysimeters is deduced from the total weight of the lysimeter by considering that the mass of the dry soil is constant and by neglecting the vegetation mass compared to the mass of the 2 m soil and water. Because we have



no measurement of the weight of the dry soil, we arbitrarily consider that the initial observed and simulated total water masses are equal for the BC66 experiment.

Time evolutions of the masses observed and simulated by BC66, VG80 and BCVG are presented in Fig. 3 for the seven lysimeters at 1-hour time step. Fig. 4 presents the statistical scores: normalized root mean square error ($NRMSE$) and the mean bias. With soil parameters determined in situ, the evolution of the total soil water mass is well reproduced by ISBA and differences between the three simulations are not pronounced, especially for the $NRMSE$ (Fig. 4). Results are better for OPE experimental station lysimeters with small $NRMSE$ (below 0.56 kg.kg$^{-1}$). Bias is more important for lysimeter G3, where thick alfalfa is present. A comparison of the three experiments shows that BCVG experiment obtained a better score with mean $NRMSE$ of 0.478 kg.kg$^{-1}$ (0.53 and 0.55 for BC66 and VG80), average biases of 18.1 kg (19.18 and 21 for BC66 and VG80), and larger R$^2$ (not shown). BCVG experiment obtains better scores than the other experiments in more than 42 % of cases.

### 4.1.2 Water Content

Time evolutions of the water content at 0.5m are shown Fig. 5 while the scores are presented for the 2 depths for which there are the most usable observations, i.e 0.5 m and 1 m (Fig. 4). For lysimeters with vegetation (G3, G4, O1, O2, and O3), the roots draw water in the summer period which reduces the volumetric water content, causing a more pronounced contrast in volumetric water content between winter and summer than for bare soil lysimeters. This behavior is well represented by ISBA: biases are negligible (<6. $10^{-2}$ m$^3$.m$^{-3}$) and dynamics are correct ($R^2 >0.5$, not shown), although O1 obtained the worst bias at 0.5 and 1 m. Still minor differences between experiments BC66 and BCVG appear. VG80 obtains weaker statistical scores in 96 % of the cases, because soil water saturation is reached too rapidly.

It should be noticed that agreement between observed and simulated soil volumetric water contents is weaker at the shallow depth available only at OPE (0.2 m) than at the other depths with NRSME$\geq$ 0.8 m$^3$.m$^{-3}$ and R$^2$ < 0.6 (not shown). This can be explained by the different processes which can modify the structure of the soil at the surface: the intensity of precipitation can increase the soil surface sealing (Liu et al., 2011; Assouline, 2004), and the soil heterogeneity can increase in response to plant or biological activities (Brown et al., 2000; Beven and Germann, 1982). Such processes are not represented in ISBA.

### 4.1.3 Drainage

Drainage at 2 m depth is measured at an hourly frequency. However, to compensate for the measurement limits associated with a 0.1 m.h$^{-1}$ threshold, the data is aggregated daily. The annual volume drained varies significantly between lysimeters of the same sites (Table 2), although it is assumed they are exposed to the same atmospheric conditions. On the GISFI experimental station, the mean annual drained water is maximum on bare soil lysimeters G1 and G2 (>300 mm.year$^{-1}$). The mean annual drained water is two to three times lower for the lysimeters covered by vegetation. At the OPE experimental station, all the lysimeters have a vegetation cover, and the mean annual drained water shows a variation of only 16 % from 300 to 363





mm.year$^{-1}$. Such amounts are comparable to the volume drained on the bare soil of the GISFI experimental station, which is mainly due to higher mean annual precipitation at OPE.

Daily mean annual cycles of drainage are shown in Fig. 6. At the GISFI experimental station, drainage occurs almost all

year long for bare soil lysimeters G1 and G2. The well developed vegetation cover in G3 causes a decrease in both drainage intensity and drainage duration. At the OPE experimental station, the drainage occurs mainly from October to June (May if the year 2016 is excluded, as a large rainfall event occurred in May-June 2016) , with similar cycles for the 3 lysimeters. The annual cycle is well simulated, with more discrepancies for the VG80 experiment, which tends to overestimate the drainage period.

Daily drainage is shown in Fig. 7 and the scores are given in Fig. 4. Even if the annual volume drained is higher at the OPE lysimeters, the maximum drainage intensities over the observed period are similar at both sites: they vary between 27.4 and 34 mm.day$^{-1}$ at OPE experimental station and between 24 and 33 mm.day$^{-1}$ at the GISFI site.

The three experiments reproduce the daily drainage with relatively low biases ($<0.5$ mm.day$^{-1}$), worst biases being ob-

tained by VG80 especially on the GISFI lysimeters. Dynamics are also well reproduced, with a higher correlation coefficient obtained for BCVG experiment (not shown). The Nash-Sutcliffe efficiency (NSE, Nash and Sutcliffe (1970)) shows better scores for experiment BCVG with an average of 0.55 (0.52 and 0.41 for experiments BC66 and BCVG, respectively).

### 4.2 Water Budget

Lysimeters give access to an estimate of the actual evapotranspiration (Schrader et al., 2013; Gebler et al., 2015). Neglecting

lateral runoff (not present in these lysimeters), the following simple water balance equation allows to estimate annual evapotranspiration ($E$) from annual precipitation ($P$), drainage ($Q_{drain}$) and a water mass variation negligible ($\Delta W$) for each lysimeter:

$$E = P - Q_{drain} - \Delta W \tag{8}$$

Fig. 8 presents the water budget observed and simulated over the entire period for all lysimeters. The total evapotranspiration

and the drainage ratios to the total precipitation are expressed in percentage. At the GISFI experimental station, between 50 and 80 % of the annual rainfall is evapotranspired, with maximal uptake on lysimeter G3 with the densest vegetation cover. At the OPE experimental station, evapotranspiration corresponds to nearly 50 % of the rainfall.

The annual budgets estimated by ISBA are rather close to the observations, but the evapotranspiration is underestimated on all lysimeters except G2 O2, and O3 for experiments BC66 and BCVG. VG80 experiment largely underestimates evapotranspira-

tion on G1, G2 and G4. The absolute bias averaged over all the lysimeters is lower for BC66 and BCVG experiments (5.3 %) compared to VG80 experiment (8.6 %).





### 4.3 Intense Drainage events

In the previous sections (4.1.3, 4.2), drainage was analyzed on complete chronicles, where strong daily drainage events were detected. In order to check the ability of the three experiments to reproduce strong soil water dynammic, a focus is made on

intense drainage events. All daily drainage fluxes larger or equal to the 99th percentile of the daily drainage distribution over the entire period are selected for each lysimeter. These Q99 values are higher for OPE experimental station lysimeters ($>13$ mm.day$^{-1}$) than for GISFI experimental station lysimeters (5.4 to 9 mm.day$^{-1}$). A total of 110 events on the set of the 7 lysimeters is selected. 75 % of these intense drainage events appear from October to March, i.e. during the wet period when the soil is near saturation. The remainder occurs in May and June associated with intense precipitation events as generally

observed in this region.

Fig. 9 presents winter intense drainage events in February 2016 for two contrasted lysimeters (O1 with vegetation and G2 with bare soil). February 2016 corresponds to a period of approximately one month (31 days for O1 and 27 days for G2), with some daily precipitations above 20 mm.day$^{-1}$, and an initially wet soil. Fig. 10 presents late spring events in June 2016

that led to a flood event of the Seine river (Philip et al., 2018). This event lead to large accumulated precipitation (166 mm at OPE and 210 mm at GISFI in 10 days) and intense daily precipitation with a maximum on May, 30th that reached above 40 mm.day$^{-1}$ and 70 mm.day$^{-1}$ at the OPE and GISFI experimental stations, respectively. Fig. 9 and 10 present observed daily precipitation, hourly observed and simulated soil profile saturation, and daily observed and simulated drainage.

VG80 tends to simulate a too wet soil compared to the observations, especially for O1 lysimeter and in depth for G2 lysimeter. On these lysimeters, BC66 and BCVG reproduce well soil moisture profiles. The drainage simulated by VG80 is occurring too early with an underestimation of the maximum drainage intensity during the period. These weaknesses are consistent with the simulation of a too wet soil. Conversely, the drainage dynamic of these events are well reproduced in phase and maximum intensity by BC66, and even better by BCVG as highlighted by the good NSE scores.


When the same comparison is made on the 110 selected intense drainage events, the scores are significantly better for BCVG. It exhibits the lowest bias (1.11 mm.day$^{-1}$) compared to the other experiments (1.3 mm.day$^{-1}$ for BC66 and 1.26 mm.day$^{-1}$ for VG80), as well as the higher NSE criteria (0.76 for BCVG compared to 0.69 for BC66 and 0.61 for VG80).

### 4.4 Synthesis

To summarize the results, Taylor diagrams are used to quantify the degree of correspondence between the modeled and observed behavior in terms of three statistics: the Pearson correlation coefficient, the root-mean-square error, and the normalized standard deviation (Fig. 11). These scores are computed using all the seven lysimeter time series, with a single result for BC66, VG80 and BCVG experiments.




For water mass and volumetric water content (Fig. 11a and b), scores are calculated at an hourly time-step, while a daily time step is used for the drainage over both the full period and the 110 drainage intense events (Fig. 11 c and d). Experiments BC66 and BCVG both obtain good results, especially to predict water mass and volumetric water content. However, the BCVG experiment obtains better results for drainage, in particular during intense events. Consistently with previous results, the VG80 experiment obtains significantly lower scores, notably for drainage.

## 5 Sensitivity Experiments

### 5.1 Homogeneous soil profile

LSMs commonly use a homogeneous soil profile. To evaluate the influence of the variation with depth of soil hydrodynamic parameters on our simulations we performed a sensitivity experiment with uniform soil profile in the combined BCVG configuration ($BCVG_{HOM}$). This uniform profile is fed with the vertical mean value of each parameter for each lysimeter (cross on Fig. 2). In terms of water budget (Fig. 8), compared to BCVG, the homogeneous profile in $BCVG_{HOM}$ increases drainage at the bottom and reduces the evapotranspiration (except in G4), because periods of drainage are longer especially during the recession. $BCVG_{HOM}$ degrades the scores in terms of water content and drainage (Fig. 11). The focus on the two intense drainage periods (Fig. 12) show that $BCVG_{HOM}$ underestimates the high water content in the deep soil layer compared to observations and reference simulations. This bias induces a delay on the drainage as shown on Fig. 12 for the O1 lysimeter.

To determine if the vertical profile of one single parameter has more impact on the simulations, we performed additional experiments with homogeneous soil profiles for all parameters except for one that keeps its estimated heterogeneous profile. These experiments performed with either $n$, $\omega_{sat}$, $\psi_{sat}$, or $k_{sat}$, demonstrate the importance of $n$ (and therefore b), especially to simulate well the drainage dynamic and intense drainage events. Using the seven lysimeter complete drainage time series as in section 4.4 (or only intense drainage events), Table 3 shows that $n$ is the most important parameter with a NSE score equal to 0.38 (0.49, respectively), with slightly lower NSE scores that the BCVG reference experiment (0.55 and 0.76, respectively). The second most important parameter is $\psi_{sat}$ with a NSE score of 0.33 (0.39), while other parameters do not exceed 0.30 (0.31). These results are in agreement with previous studies (Ritter et al., 2003) that demonstrated a strong sensitivity to $n$ and low sensitivity to $k_{sat}$.

### 5.2 Usual pedotransfer functions

LSMs commonly use PTF to derive soil hydrodynamic parameters from soil textural information. As shown in Fig. 2, the soil parameters estimated from our measurements can be very different from those derived from six usual PTF (Clapp and Hornberger, 1978; Cosby et al., 1984; Carsel and Parrish, 1988; Wosten and van Genuchten, 1988; Vereecken et al., 1989; Weynants et al., 2009). This is especially true for the parameter $n$ (or $b$). To investigate the impacts of such differences, a sensitivity experiment noted $BCVG_{PTF}$ is performed in which the soil hydrodynamic parameters in each soil horizon are derived from the





365     mean of these PTFs. In terms of water budget (Fig. 8), $BCVG_{PTF}$ increases drainage and reduces evapotranspiration compared to BCVG (except for G4, O1 and O3), because periods of drainage are longer especially during the recession (as previously for homogeneous profiles). The skill scores are degraded (Fig. 11) with correlation lower than $0.4$ for drainage. In February and June 2016, the soil water profile simulated by $BCVG_{PTF}$ is strongly underestimated compared to observations (Fig. 12). This weakness induces a significant delay of the simulated drainage compared to observations and to other experiments.

370     Once again, the $n$ (or $b$) parameter seems to be the key to this weakness. Indeed, we performed additional experiments using the $BCVG_{PTF}$ configuration except for one parameter (either $n$, $\omega_{sat}$, $\psi_{sat}$, or $k_{sat}$) that keeps its estimated in situ value. As previously seen (section 5.1), the use of the in situ estimated values of $n$ (or $b$) drastically improves scores compared to $BCVG_{PTF}$ (Fig. 11). This is also the case for the simulation of the water budget (Fig. 8), the soil water profile (Fig. 12), and especially the soil drainage dynamic (Table 3 and Fig. 12).

## 375  6   Discussion and Conclusion

This study uses time series (up to 7 years) from several lysimeters to evaluate the dynamics of water transfer in the unsaturated zone simulated with the land surface model ISBA. These observations allow to estimate heterogeneous soil parameter profiles. Although the original version of the ISBA performed well, a new set of water closure relationships, which estimate the evolution of soil properties with soil moisture are tested. The comparison of the three relationships shows that, when soil parameters 380 and meteorologic forcing are known, ISBA reproduces the evolution of soil hydrology and vegetation processes reasonably well.

    The experiment using the VG80 water closure relationships exhibits more difficulty reproducing the drainage dynamic, in particular during intense drainage events, than the original ISBA version using the BC66 equations. It is partly linked to the 385 complexity of the VG80 hydraulic conductivity function. The $l$ parameter in equation 5 is difficult to set even with direct observation. In our study, it is fixed at the classical 0.5 value for some lysimeters, but can vary drastically in others consistently with the literature (Wosten and van Genuchten, 1988; Wösten et al., 2001; Schaap et al., 2001), underlying the difficulty to estimate this parameter for regional to global scale applications. In addition this hydraulic conductivity function is not numerically stable for small values of $n$ ($n < 1.09$). The combination of the soil water curve from VG80 and the hydraulic conductivity from 390 BC66 solves these problems and even improves the simulation of the soil drainage dynamic.

    Our observations show that the soil hydrodynamic parameters in each lysimeter are strongly heterogeneous with depth, while LSMs generally use homogeneous profiles. Using additional experiments with such homogeneous profiles, we found that even if the simulated soil water and drainage dynamics remain acceptable compared to the observations, all the skill scores 395 are worsen compared to the experiments with an heterogeneous profile. This finding support the need to account for vertical heterogeneity of soil hydrodynamic parameters (King et al., 1999; Mirus, 2015; Hengl et al., 2017; Vogel, 2019; Fatichi et al., 2020; Bauser et al., 2020; Gebler et al., 2017) to improve the simulation of soil water and drainage dynamics (Stieglitz et al.,

1997; Mohanty and Zhu, 2007; Decharme et al., 2011; Vereecken et al., 2019). This is a challenge to simulate groundwater recharge on regional and global scales.


We also found that the parameter $n$ (or $b$), which characterizes the shape of the soil-water retention curve, estimated from our observations significantly differ from those derived from PTF commonly used in LSMs (Clapp and Hornberger, 1978; Cosby et al., 1984; Carsel and Parrish, 1988; Wosten and van Genuchten, 1988; Vereecken et al., 1989; Weynants et al., 2009). Some additional experiments suggest that the values of $n$ (or $b$) derived from usual PTF are not suitable to simulate the drainage

dynamic, at least over the 7 lysimeters used in this study. In addition, this parameter exhibits the largest heterogeneity with soil depth. Neglecting this behavior contributes to degrade the simulated drainage dynamic. Note that this heterogeneous behavior of $n$ (or $b$) is still under consideration in the literature. As in our study, some authors observed a decrease of $n$ with soil depth (Ritter et al., 2003; Jhorar et al., 2004; Schwärzel et al., 2006), while some others showed an increase (Groh et al., 2018) or even no change (Schneider et al., 2021). In any case, $n$ (or $b$) could be the key parameter to correctly simulate drainage dynamic

and groundwater recharge with LSMs.

We recognise that this last assumption has to be confirmed over many other experimental field sites. Indeed, this study is based on two experimental sites with similar climatic conditions, with low intensity precipitation events compared to other regions. It would be interesting to conduct additional studies in other contrasting climates.


Finally, this study increases the confidence that LSMs are powerful tools to simulate the recharge of groundwater, in different environmental conditions, with many soils and vegetation covers, and therefore can be used for many applications in hydrology at both the regional and the global scales.

*Author contributions.*  The article was written by AS with contributions from all co-authors. AS, BC, and FH developed the study design. NE, POR, and PFC collected the lysimeter data and assisted in their use. AS analyzed the data, developed new parameterizations, and ran the models with general support from BD and FH, and specific support from CD in parameterizing vegetation and PLM to weather forcing.

*Competing interests.*  The authors declare that they have no conflict of interest.

*Acknowledgements.*  The authors are very grateful to Corinne Leyval from LIEC Nancy and Geoffroy Séré from LSE for all the useful

discussions and their willingness to provide access to the data sets from GISFI. The authors thank the GISFI for providing field site data at the experimental station in Homécourt and in particular the technical staff, Cyrielle Boone. The OPE dataset was provided by ANDRA with the help of Catherine Galy from ANDRA. Thanks to the IR OZCAR infrastructure for enabling the connection between research from





different topics at the various sites. The primary author is funded by the "Centre National de Recherches Météorologiques" of Météo-France, the "Agence de l'Eau Seine Normandie" and the "Laboratoire de Géologie de l'Ecole Normale Supérieure" of Paris.



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





**Table 1.** Description of lysimeters: soil type, vegetation cover, number of texture observations, and textures (in % of clay and sand) at different depths. HyCa stands for Hypereutric Cambisol

| Site | GISFI experimental station | | | | OPE experimental station | | |
|---|---|---|---|---|---|---|---|
| Lysimeters | G1 | G2 | G3 | G4 | O1 | O2 | O3 |
| Soil | Technosol | Technosol | Technosol | Cambisol | HyCa | Cambisol | HyCa |
| Soil cover | bare soil | bare soil | Alfalfa | Grass | Grass | Grass | Grass |
| Layers | 1 | 1 | 1 | 4 | 6 | 1 | 6 |
| (Sand,Clay) (%,%) | | | | | | | |
| Homogeneous | (61.6,14.3) | (61.6, 14.3) | (62.4,15.2) | (32,25) | (3,36) | (31,41) | (18,47) |
| 0.2m | "" | "" | "" | (20,15) | (11,4) | (50.4,18) | (24,28) |
| 0.5m | "" | "" | "" | (17,26) | (0, 67) | (42,27) | (16,53) |
| 1m | "" | "" | "" | (34,33) | (0, 19) | (22,6) | (16,53) |
| 1.5m | "" | "" | "" | (56,24) | (0, 19) | (22,6) | (16,53) |





**Table 2.** Description of the available observations for each lysimeter: observation period, mean Annual precipitation (Precip) and drainage (Drain). For each type of data, the available depths are indicated (cm). Quality of measurements is given as percentage of missing data: meteo gap for the meteorological forcing, defect for the lysimeters measurements.

| Site | GISFI experimental station | | | | OPE experimental station | | |
|---|---|---|---|---|---|---|---|
| Lysimeters | G1 | G2 | G3 | G4 | O1 | O2 | O3 |
| Period | 2011-2016 | 2009-2016 | | 2011-2016 | 2014-2019 | | |
| Precip (mm.year$^{-1}$) | 727 | | | | 876 | | |
| Drain (mm.year$^{-1}$) | 317 | 337 | 115 | 170 | 312 | 304 | 363 |
| Total Water mass | full column | | | | full column | | |
| Volumetric water content | 100-150 | 50-100-150 | 50-100-150 | 50 | 20-50-100-150 | | |
| Matric potential | 100-150 | 50-100-150 | | 50 | 20-50-100 | | |
| Drainage | 200 | | | | 200 | | |
| Temperature | 50-100-150 | | | | 20-50-100-150 | | |
| **Quality of data** | | | | | | | |
| Meteo gap (%) | 12 | | | | 10 | | |
| Defect (%) | 16 | 8 | 23 | 0 | 0 | | |





**Table 3.** Nash Sutcliffe efficiency (NSE) for the 7 lysimeters drainage simulations ($Q_{drain}$), and during intense drainage events ($Q_{int}$), for experiments with soil hydrodynamic parameters set with an homogeneous vertical profile ($BCVG_{HOM}$) or computed using usual PTF ($BCVG_{PTF}$), or derived from observation (BCVG). The NSE of the additional experiments with one parameter ($n$, $\psi_{sat}$, $\omega_{sat}$, or $k_{sat}$) that keeps the reference values are also given.

| NSE | $BCVG_{HOM}$ | | $BCVG_{PTF}$ | | Ref. BCVG | |
|---|---|---|---|---|---|---|
| | $Q_{drain}$ | $Q_{int}$ | $Q_{drain}$ | $Q_{int}$ | $Q_{drain}$ | $Q_{int}$ |
| | 0.28 | 0.31 | 0.02 | -0.5 | 0.55 | 0.76 |
| $n$ | 0.38 | 0.49 | 0.38 | 0.51 | | |
| $\psi_{sat}$ | 0.33 | 0.39 | -0.1 | -0.5 | | |
| $\omega_{sat}$ | 0.28 | 0.3 | -0.05 | -0.41 | | |
| $k_{sat}$ | 0.3 | 0.29 | -0.01 | -0.47 | | |

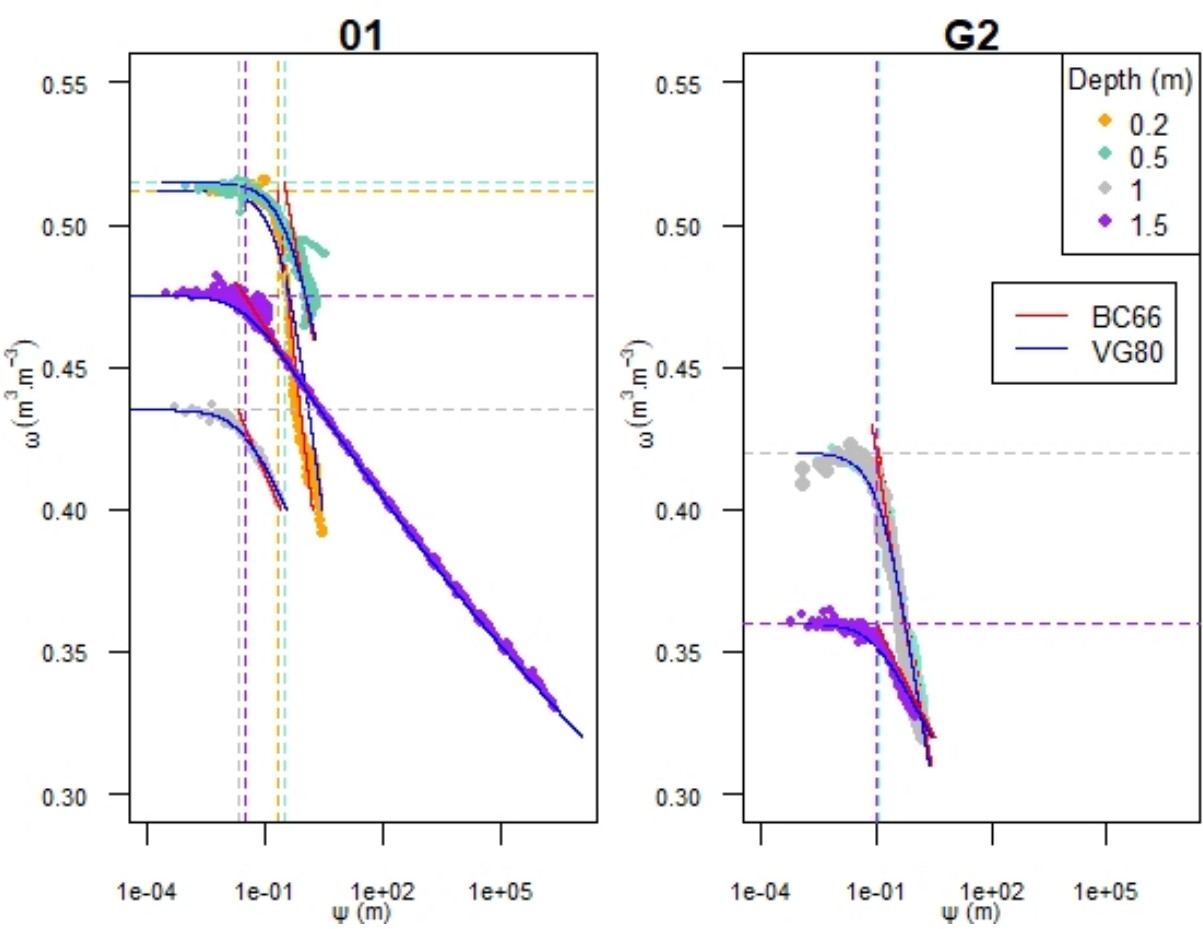

**Figure 1.** Relationships between volumetric water content ($\omega$) and logarithm of the absolute value of the soil matric potential ($\psi$) for lysimeters O1 and G2. Observations at 0.2, 0.5, 1 and 1.5 m depth are in dot (orange, aquamarine, grey and purple respectively), estimations are in red and blue for BC66 and VG80 experiments, respectively.







**Figure 2.** from left to right and top to bottom: Hydraulic conductivity at saturation ($K_{sat}(10^6 m/s)$), volumetric water content at saturation ($\omega_{sat}(m^3.m^{-3})$ ), $b$ and $n$, matric potential at saturation ($\psi_{sat}(m)$ ) and alpha ($m$). Estimations from in situ measurements are represented by triangles at 0.2, 0.5, 1 and 1.5 m (orange, aquamarine, grey and purple, respectively), and their mean (homogeneous) values are represented by a star. The values derived from six pedotransfer function are shown by a boxplot presenting the median, 25 % and 75 % quantiles.



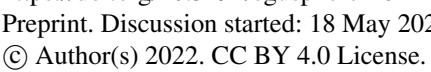

**Figure 3.** Hourly time series of the total water mass ($kg$) from GISFI (G1, G2, G3, G4) and OPE (O1, O2, O3) lysimeters. Observations are in black, the BC66 experiment in red, VG80 in blue, and BCVG in green. The grey shaded areas correspond to periods when the meteorological forcing is reconstructed and the blue shaded areas to the periods when data is of low quality.







**Figure 4.** Statistical scores on daily chronicles reached by the BC66 (red), VG80 (blue) and BCVG (green) experiments. The mean bias versus $NRMSE$ for the total water mass (top) and for the volumetric water content (medium) at 0.5 m ($\omega_{50}$) and 1 m ($\omega_{100}$) depth are shown. The mean bias versus the Nash Sutcliffe efficiency for drainage flux are shown in the bottom. Each lysimeter is represented by its identifier. For each experiment, ellipses represent the multivariate Student's t-distribution following (Fisher, 1992) .



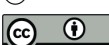

**Figure 5.** Same as figure 3 but for hourly volumetric water content (m$^3$.m$^{-3}$) at 0.5 m depth





**Figure 6.** Daily mean annual cycles of the drainage time series (mm.day$^{-1}$) computed as the average of all available data for each day of the year over the entire period with data for each lysimeter. Observations are in black, simulations in color







**Figure 7.** Same as figure 3 but for daily drainage (mm.day$^{-1}$).



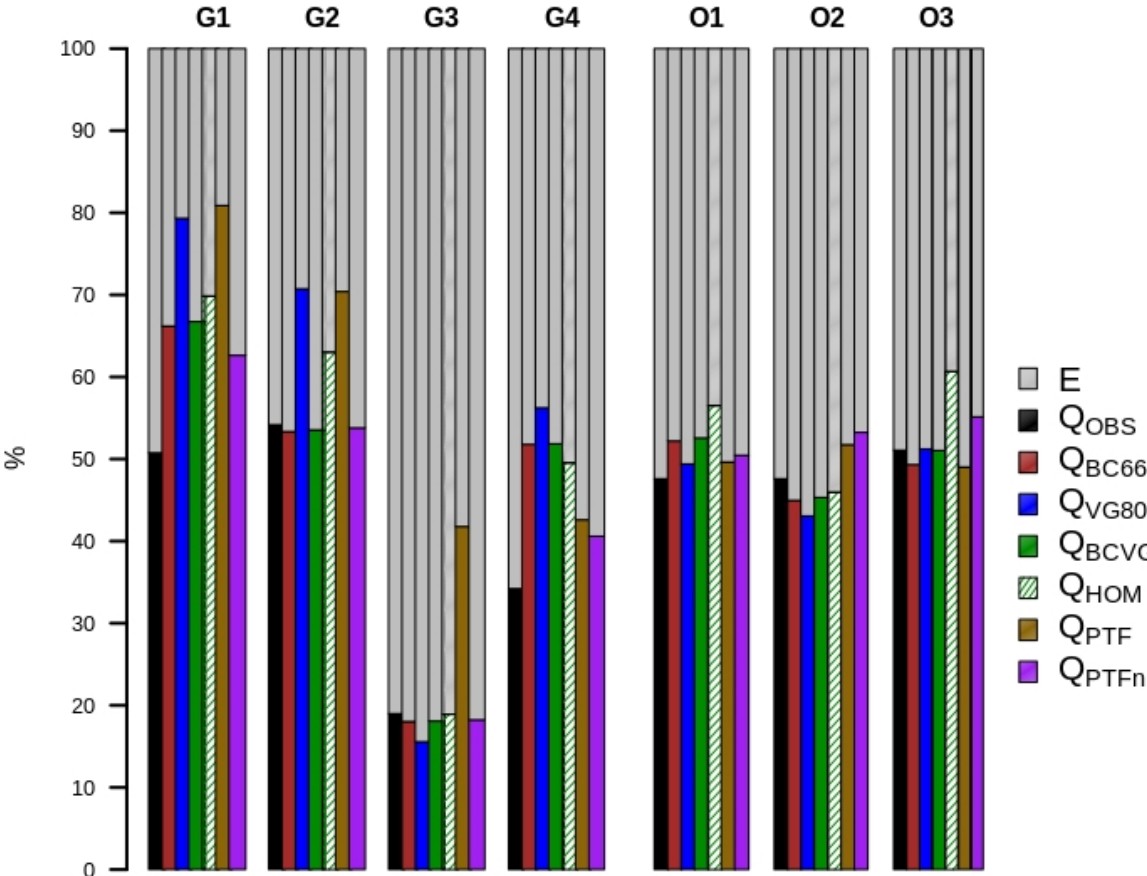

**Figure 8.** Water budget partition of the precipitation into drainage and evapotranspiration expressed in % for lysimeters of GISFI (G1, G2, G3, G4) and OPE (O1, O2, O3) for each experiment. Evapotranspiration (E) is in grey for observations and for each experiment. The observed drainage ($Q_{obs}$) is in black, the BC66 simulated drainage ($Q_{BC66}$) in red, $Q_{VG80}$ in blue, and $Q_{BCVG}$ in green. Drainage simulated by additional experiments with an homogeneous soil profile ($Q_{HOM}$) is in green hatched, with parameters estimated from usual PTFs ($Q_{PTF}$) in brown, and with parameters estimated with usual PTFs except $n$ estimate in situ ($Q_{PTFn}$) in purple. These last experiments are defined in section 5.





**Figure 9.** Daily precipitation ($mm.day^{-1}$), hourly effective wetting saturation profile (%) observed (OBS) and simulated by BC66, VG80 and BCVG, and daily drainage ($mm.day^{-1}$) observed (in black), and simulated by BC66 in red, VG80 in blue and BCVG in green during intense drainage in February 2016 for lysimeters O1 and G2. The Nash Sutcliffe efficiency (NSE) for each simulated drainage is also given.





**Figure 10.** Same as figure 9 but for June 2016.

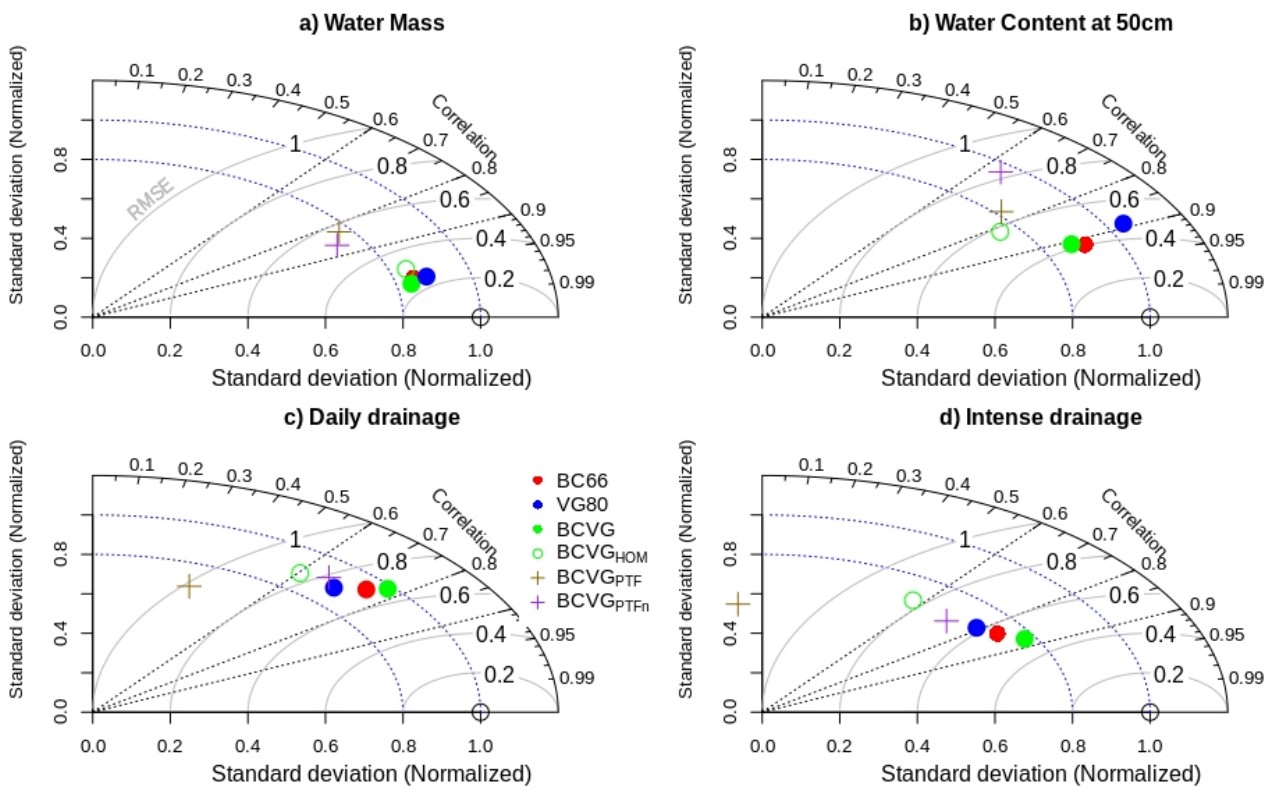

**Figure 11.** Taylor diagrams for hourly Total Water Mass (a), hourly Volumetric water content at 0.5m depth (b), Daily drainage (c) and Daily intense drainage events (d). Experiment BC66 is plotted in red, VG80 in blue and BCVG in green. Additional experiments with an homogeneous soil profile ($BCVG_{HOM}$) are represented as an open green circle, with parameters estimated from usual PTFs ($BCVG_{PTF}$) by a brown cross, and with parameters estimated with PTF except $n$ estimated in situ ($BCVG_{PTFn}$) by a purple cross. The Pearson correlation coefficient, the root-mean-square error (RMSE), and the normalized standard deviation are summarized in this diagram.



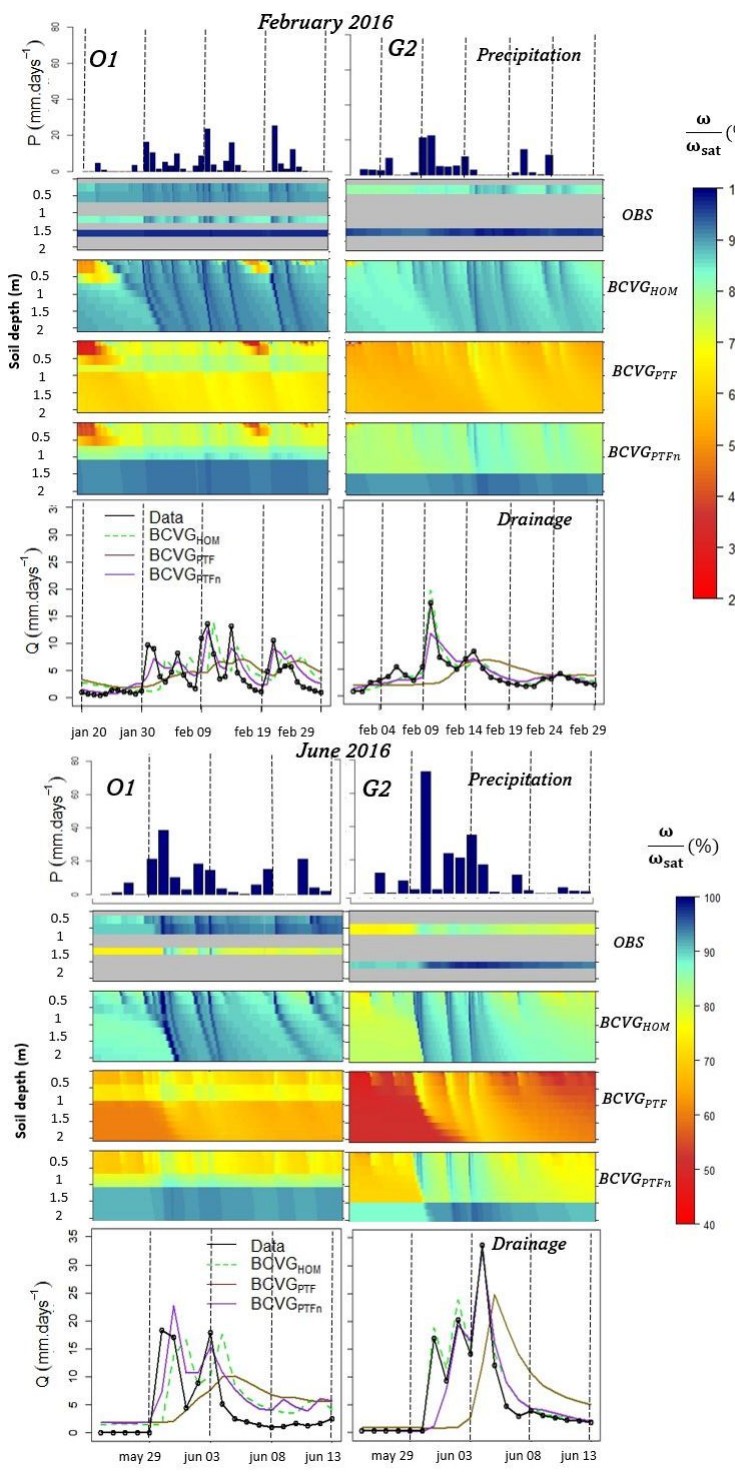

**Figure 12.** Same as figures 9 and 10 but for sensitivity experiments with homogeneous soil profile ($\mathrm{BCVG}_{HOM}$), with soil parameters from the usual PTFs ($\mathrm{BCVG}_{PTF}$), and with parameters from the usual PTFs except for $n$ estimated in situ ($\mathrm{BCVG}_{PTFn}$).