# Peer review of "Improvement of the soil drainage simulation based on observations from lysimeters"

_EGUsphere, 2022_

## Referee Comment (RC3)

**Review of Improvement of the soil drainage simulation based on observations from lysimeters (https://doi.org/10.5194/egusphere-2022-274)**

The goal of this study is to use a LSM to reproduce soil water mass, volumetric water content and drainage flux observed in seven lysimeters during a period of more than five years. The simulations are performed with three different parametrizations for the soil hydraulic properties, namely the common functions of (i) Brooks and Corey (1966) [BC66], (ii) van Genuchten-Mualem (1980) [VG80], and additionally (iii) a previously proposed "hybrid" approach, where the VG80 retention function is combined with the BC66 conductivity function [VGBC]. Considering standard diagnostic variables, the authors find a best performance with the VGBC approach and a worst performance for the VG80 approach. Ancillary studies further investigated the replacement of heterogeneous, layered soils with uniform properties. The authors note that all the results deteriorate compared to the tests with a heterogeneous profile and conclude that the vertical heterogeneity of the soil hydrodynamic parameters must be taken into account. Finally, the use of PTF-derived hydraulic properties in the simulation is compared to the use of parameters based on in-situ measurements of water content and matrix potential. Basically, the in-situ derived VG80 parameters n differ strongly from the PTF-derived values, leading to significantly different simulation results, which are worse for the PTF-based simulations.

**General Comment**

Using lysimeter data in soil hydrologic analyses is always exciting because it shows us what we do and do not understand about soil hydrologic processes. And it gives us clues about the reliability or unreliability of local soil moisture sensor readings. So this work is a very good exercise, and the use of lysimeter data for this type of study has merit. Furthermore, the paper is well written and a pleasure to read. However, I have a problem with the main idea of the paper, which is to propose the use of a VGBC parameterization in soil hydrology.

The approach taken by the authors here may be nice from a numerical point of view, but from a physical point of view it must be considered unsuitable. The reason, in short, is that a smooth, continuously derivable soil water retention function (WRC) is combined with an incompatible shaped hydraulic conductivity function (HCC). This does not make physical sense. Furthermore, the comparison of the three functional approaches is most likely biased by problems with the VG80 model when it is used to parameterize the WRC of a fine-grained soil or a soil with a broad pore size distribution (indicated by a parameter value of $n$ close to 1). In such cases, the VG80 hydraulic conductivity curve (HCC) exhibits an abrupt drop near saturation (Durner, 1994). This drop (i) leads to a severe underestimation of the hydraulic conductivity when the prediction of HCC is based on the measured saturated hydraulic conductivity (Vogel et al., 2001) and (ii) negatively affects the performance of the numerical solvers, i.e., their stability and accuracy during the transition between saturated and unsaturated conditions (Ippisch et al., 2006). This could be the reason why the authors of this study limit the smallest value of n to a threshold value (arbitrarily chosen for numerical reasons) of n > 1.09 and also find in the performance comparison that the VG80 model leads to overly wet soil profiles.

To eliminate the well-known artifact of the VG80 conductivity curve for soils with small $n$ three main approaches have been proposed in the past. They are (i) shifting the entire pore-size distribution by an air-entry value (Kosugi, 1994), (ii) introducing an explicit air-entry pressure into the van Genuchten model (Vogel and Cislerova, 1988; Vogel et al., 2001; Ippisch et al., 2006), or (iii) truncating the pore-size distribution (Malama and Kuhlmann, 2015; Iden et al., 2015). Some other workarounds have also been proposed (e.g. Schap and van Genuchten, 2006). The second approach is historically the most commonly used. It effectively eliminates the drop in HCC near saturation and the associated negative effects on the behavior of the numerical solvers of the Richards equation. The drawback, however, is that the resulting WRC is no longer continuously differentiable, i.e., the soil water capacity function becomes discontinuous, which in turn can cause problems with numerical solvers of the Richards equation in some situations.

The VGBC approach adopted by the authors in their study is basically a variant of approach iii, i.e., they leave the RETC in its original smooth form, but limit the conductivity to a constant maximum value reached at an "air entry" value (somehow arbitrarily constructed for soils with small n values).

It is certainly fair to try such an approach. However, the paper fails to convince me that this approach is worth pursuing. An important reason for this opinion is the fact that the results in the paper are not fully comprehensible to the reader, since crucial information about the modeling scenario and especially the hydraulic properties of the soil is missing. So we have to believe the results or not. In addition, the paper has shortcomings, especially in the methods section of soil hydrology (Section 3.1), as will be presented later in this review. As is so often the case in soil hydrology, the devil is in the details, and therefore I recommend reviewing the study again in detail, taking into account the comments below. Specifically, I recommend repeating the analysis (or adding a scenario to the analysis) with modified BC80 HCC functions the remove the abovementioned VG80 artifact. This is probably most straightforward using the HCC functions of Iden et al. (2015, Eq. 18 therein). If desired, I [Wolfgang Durner] would be willing to provide the authors with the respective HCC functions for their lysimeter's retention curves.

**SPECIFIC COMMENTS**

1. Basic approach:
   As mentioned before, I believe that a mixture of VG RETC and BC HCC is not a good solution to solve the problems with VG80, even though others have tried it and it leads to smooth simulations. The reason is that this is simply unphysical. Above the air entry value, we have significantly increasing water contents, but constant conductivity. This contradicts the soil hydrological evidence.

2. Equations:
   The equations of the model used in the study must be either directly stated or clearly referenced. A statement such as "A complete description of the model equations used to simulate water transport can be found in xxxx (xxxx)" would be appropriate, especially for the soil hydrology processes that are the core of the paper. I have not found this. I understand that the mixed-form Richards equation was used, but with what kind of boundary conditions? This is especially of interest for the lower boundary condition, where Figs. 9, 10 and 12 indicate drainage under apparently unsaturated conditions, which is not possible when the lysimeters are operated with the traditional lysimeter boundary condition (i.e., seepage).

3. Data:
   Which sensors were used? Since in science the reproducibility of results is of utmost importance, the high quality journals nowadays require that all necessary data are given. This bloats the manuscript itself too much, but it is possible to either put the data in data archives or publish them separately in appropriate data journals or attach them directly to articles as additional information. As a specific example, I would actually be interested in repeating the June 2016 G2 lysimeter scenario because the sudden breakthrough of water to the drainage seems difficult to explain with the VG80 data provided. However, with the information currently available, it is not possible to perform such a test and judge the validity of the scientific statements.

4. Results.
   Some of the results are shown as examples, which is fine. However, the reader should have access to the full results. So please use the supplementary material to list data not shown in the manuscript! For me, especially diagrams like figure 1 for lysimeters O1 and G2 are of great interest, also for the five other lysimeters.

5. Lower boundary condition:
   In line 181 authors state "*By assuming the unit-gradient assumption, the drainage at 2 m can be considered equal to the hydraulic conductivity. ksat is then assumed equal to the observed drainage when the soil water content is saturated*" --- The statement is correct if really unit-gradient conditions prevail in the lysimeter. In typical lysimeters, where the water flows freely into the atmosphere, this

would require full saturation of the entire lysimeter(!). In suction controlled lysimeters it is different, but I did not find a statement about this in the paper. Do you have any indication that this assumption can be (approximately) true if the whole lysimeter is not saturated? Or any reference where the validity of this approximation has been shown?

6. Conductivity functions.
Conductivity functions are key to the outcome of the simulations. As always with simulations, we can say, "What you put in determines what you get out." I would like to see the functions used in this comparison in appropriate graphs. The single Figure 1 is difficult to read and not sufficient. Please see my comments on Figure 1, below. Also, the coefficients of the hydraulic functions used in the simulation study should be listed in a table (perhaps in the supplemental material), as the information in Figure 2 is not sufficient to reproduce the functions.

**COMMENTS RELLTED TO SPECIFIC TEXT PASSAGES**

1. INTRODUCTION
- line 45. "*Vereecken et al. (2019) suggested a number of directions for improvement: introduce more physical processes such [... ] improve the representation of [...] soil parameters,*" --- Agreed. But I do not really see the VGBC approach as an improvement of soil parameterizations in the above-mentioned sense.
- line 56 "*These relationships are simple to parameterize and very stable numerically.*" --- There is some irony in the fact that Rien van Genuchten actually developed his parametrization to solve the problem of a discontinuous water capacity function that causes numerical problems in simulations with the Richards equation. So, VG80 should be numerically stable except for cases with very small n.
- line 60 "*However, the VG80 relationships are less stable than BC66 for coarse-textured soil, mainly because of the complexity of the hydraulic conductivity function (Vogel et al., 2000).*" --- In fact, I am not aware of this statement in the Vogel paper. I know VG80 is problematic for coarse grained soils in the dry moisture range, but in that range there is not much difference to BC66.
- line 73 "*We derive their hydrodynamic parameters directly from observation*" --- This is certainly the key to all subsequent results. As already indicated, the documentation on this important point is not sufficiently presented in the paper.

2. EXPERIMENTAL PROTOCOL
- line 85 "*These two sites are separated by a distance of 97 km.*" --- Please state here the basic hydro-meteorological parameters: total precipitation (comes too late at the end of the section), total estimated ETp, height above sea level, mean temperature.
- line 101 "*with different layers of limestone more or less cracked*" --- uuh, taking an undisturbed lysimeter in such material is difficult. Described somewhere in literature?
- line 104 "*They are equipped with suction and temperature probes as well as time-domain reflectometry (TDR) probes*" --- please specify the used instruments/sensors.
- line 113 "*The gaps represent up to 15% of the observations for the GISFI site* (Table 2)." --- I tried to figure that out from table 2, but find there a different value, 12%.
- Line 115 "*atmospheric forcing*" – how expressed? How is the upper soil hydraulic boundary condition expressed in the model?
- line 117: "*ISBA model*" --- I would like to see some more equations related to the main hydrological processes in the soil. For example, I have not found the implementation of the boundary conditions, and perhaps the calculation of isothermal vapor conductivity via a function of soil texture could somehow be shown without having to consult Braud et al. (1995).
- line 154. As a soil hydrologist, I am somewhat surprised at this type of notation. Why psi(w) and K(psi), instead of expressing both w and K as a function of psi (which is common, at least in the soil hydrology community)? Of course, this is meant only as a comment and is not of any consequence....

- line 157: *"the more complex closed-form equations from van Genuchten (1980)"* --- No. The equations may look a bit more complicated, but VG's approach is certainly not "more complex". The fact that a simple algebraic equation is very short in one case, while it looks a bit more complicated in the other case, should not be interpreted as "complex" in 2022.
- line 162 *"α (m-1) [is] the inflection point where the slope of the soil-water retention curve (dω/dψ) reaches its maximum value,"* --- This is wrong. $1/\alpha$ is inbetween the inflection point and the air-entry value.
- *"line 163. "l [is] the Mualem (1976) dimensionless parameter that determines the shape of the hydraulic conductivity curve"* --- No, $l$ does not determine the "shape" of the conductivity curve, but rather the slope of the log K vs. log psi curve in the unsaturated range. The "shape" of the function is unaffected, log K (log psi) remains linear.

3. ESTIMATION OF PARAMETERS

- Line 173 *"The rich data sets collected by the lysimeters allow the derivation of the soil hydrodynamic parameters"*; *"For instance, Fig. 1 plots"* --- Figure 1 is central to the entire paper and its findings. I have a number of comments on it, which are included at the end of my comments here. These data must be given for all lysimeters, as they determine the outcome of the simulations and thus affect all conclusions of the paper.
- line 179: *"ωsat is determined by the 99th percentile of the observed soil volumetric water content"* --- to be sure: this is 6 years * 365 days * 24 values = > 52'000 water content data for each layer? Just make this clear.
- line 178: *"observed ψ-ω relationship at each depth"* --- I assume the depths listed in Table 2? Are the layer boundaries of the different materials in the simulation chosen to be at the mean distance between these observation depths?
- line 182 *"By assuming the unit-gradient assumption, the drainage at 2 m can be considered equal to the hydraulic conductivity."* --- Did you ever observe fully saturated unit gradient conditions in the lysimeter? Saturation just at the base is not a sufficient condition.
- line 193" *Vogel et al. (2000) determined a limit of n < 1.3 below which Eq.(5) is numerically unstable."* – Well, that's a complex issue. See remarks at the begin of the review.
- line 195. *"The l parameter in Eq. (5) from VG80 is estimated with a simple calibration via ISBA sensitivity experiments with l ranging from -5 to 5. "* --- This is incomprehensible: What exactly was done in this "simple calibration via ISBA sensitivity experiments"? Also, it is suspicious that the optimized values for the GISFI lysimeters are all at the limit of a permissible range. Furthermore, with $l$ = -5, we are outside the physically permissible range for many sets of hydraulic functions and might face even increasing conductivity in dry soil (see Peters et al., 2011).
- line 200: *"The derived parameters are presented in Fig. 2"* --- As mentioned above, the parameters (ωsat, ψsat, α, n, b, l and ksat) should be listed (additionally) in a Table, maybe in supplementary material.
- line 215 *"if the volumetric water content presents a slow decrease in summer at a given depth, it is considered that the roots have not yet reached this depth. The root depth is thus fixed at 2 m for lysimeters G3 and O2"* --- Hm, this is really a very large depth for the grass roots. Unusual.
- line 216 *"varies for lysimeters G4, O1 and O3."* --- Would be nice to see this illustrated.
- line 222: *"(not shown)"* – Why not shown? In digital format, we do not have space limits in the supplemental information!

4. RESULTS

- line 240 *"we arbitrarily consider that the initial observed and simulated total water masses are equal for the BC66 experiment"* --- ... and how do you get the initial simulated total water mass? Based on what initial conditions in the simulation? And is the initial water content of the VG80 identical to that of the BC66?
- line 249 *"BCVG experiment obtains better scores than the other experiments in more than 42 % of cases"* --- Certainly true, but the differences between the model variants seem to be insignificant. With the exception of lysimeter G4, where the BC66 parameterization leads to a different picture than the others, we can say that the variants give more or less the same results. The fact that in G3 all three

model variants overestimate the water mass might hint on a problem with the initial conditions or the soil mass.

- line 258 "*VG80 obtains weaker statistical scores in 96 % of the cases, because soil water saturation is reached too rapidly*"--- This could be an indication of an incorrect conductivity curve. As mentioned in the introductory section of this study, the use of VG80 is not acceptable for such small values of n, and this result is a strong indication of an incorrect conductivity curve. The study should be supplemented with a modified VG80 conductivity as suggested above.

**Table 1**

Please add some more key information: USDA soil type (e.g., sandy loam), monolithic or filled, bulk densities.

**Table 2**

Layout is puzzling: two columns for GISFI water content sensors, one column for the respective matric potential sensors.

**Table 2.** Description of the available observations for each lysimeter: observation period, mean Annual precipitation (Precip) and drainage (Drain). For each type of data, the available depths are indicated (cm). Quality of measurements is given as percentage of missing data: meteo gap for the meteorological forcing, defect for the lysimeters measurements.

| Site | GISFI experimental station | | | | OPE experimental station | | |
|---|---|---|---|---|---|---|---|
| Lysimeters | G1 | G2 | G3 | G4 | O1 | O2 | O3 |
| Period | 2011-2016 | 2009-2016 | | 2011-2016 | 2014-2019 | | |
| Precip (mm.year$^{-1}$) | | 727 | | | 876 | | |
| Drain (mm.year$^{-1}$) | 317 | 337 | 115 | 170 | 312 | 304 | 363 |
| Total Water mass | full column | | | | full column | | |
| Volumetric water content | 100-150 | 50-100-150 | 50-100-150 | 50 | 20-50-100-150 | | |
| Matric potential | 100-150 | 50-100-150 | | 50 | 20-50-100 | | |
| Drainage | | 200 | | | 200 | | |
| Temperature | | 50-100-150 | | | 20-50-100-150 | | |
| **Quality of data** | | | | | | | |
| Meteo gap (%) | | 12 | | | 10 | | |
| Defect (%) | 16 | 8 | 23 | 0 | 0 | | |

**Fig. 1**

[Figure]

a) Are these data from five years in-situ measurements (hourly resolution)? Why is there no hysteresis?

b) How are the depicted conductivity data derived? From the paper, I learned that Ksat is derived from a unit-gradient assumption, and relative K-function in the unsaturated range is predicted by the BC66 reps. VG80 model.

c) Why do conductivities decrease with decreasing suction (@1 m, soil G2, grey dots)

d) How are the RETC data for O1 @1.5 m derived? Such experimental data cannot be obtained this far into the unsaturated range! Moreover, a water content of 35% is impossible at a suction of 1E+5 m. Note that this suction condition I beyond oven dryness, i.e., $\omega = 0$!

e) What's the meaning of the dotted lines (I assume psi_s and $\omega$_s, but name it in the legend or caption!)

f) Why do we find such strongly different slopes of RETC in a packed lysimeter (G2) with homogeneous)!) soil? What are the bulk densities?

g) Where are the data for O2, and O3, and for G1, G3 and G4? → supplemental material.

**Fig. 2**

a) Ksat – I am not able to decipher the unit. $10^6$ m/s ???

b) The *n* values are very low, close to 1, but in this figure it is impossible to distinguish values near one. Please list the values also in a table.

**Fig. 9**

a) Why do we observe outflow in a system that does not reach saturation at the bottom? Are these suction lysimeters?

**TYPOS AND MINOR ISSUES**

- Terminology. For me it is puzzling to speak of "experiments" when actually different simulations are meant. The "experiment" to me is the physical experiment where a system is manipulated to observe a response. Perhaps "model variants" or "model approaches" is more appropriate?
- line 83: This is an international journal with many readers not familiar with french terrestrial research units: please define GISFI and OPE upon first occurrence.
- line 95: which bulk density?
- line 200: Typo "lystimeters".
- line 203: replace "deep" by "depth".
- line 243 „*average biases of 18.1 kg (19.18 and 21 for BC66 and VG80)*" --- Keep an eye on your digits: Better "18.1 kg (19.2 kg and 21.0 kg for BC66 and VG80"
- line 277 "if' ??????????
- line 304 "dynammic"

**Literature cited**

Brooks, R. H. and Corey, A. T. Properties of Porous Media Affecting Fluid Flow, Journal of the Irrigation and Drainage Division, 92, 61–88,455 https://doi.org/10.1061/JRCEA4.0000425, 1966

Durner W. Hydraulic conductivity estimation for soils with heterogeneous pore structure. Water Resour Res, 30, 211–23. doi:10.1029/93WR02676, 1994

Ippisch O., Vogel H-J., Bastian P. Validity limits for the van Genuchten–Mualem model and implications for parameter estimation and numerical simulation. Adv Water Resour, 29, 1780–9. doi:10.1016/j.advwatres.2005.12.011, 2006

Kosugi K. Three-parameter lognormal distribution model for soil water retention. Water Resour Res, 30, 891–901, 1994

Malama B., Kuhlman KL. Unsaturated Hydraulic Conductivity Models Based on Truncated Lognormal Pore-Size Distributions. Ground Water, 53, 498–502. doi:10.1111/gwat.12220, 2015

Peters, A., Durner, W., Wessolek, G. Consistent parameter constraints for soil hydraulic functions. Advances in Water Resources, 34(10), 1352-1365, 2011

Schaap, M. G., Van Genuchten, M.T. (2006). A modified Mualem–van Genuchten formulation for improved description of the hydraulic conductivity near saturation. Vadose Zone Journal, 5(1), 27-34.

van Genuchten M.T. A closed-form equation for predicting the hydraulic conductivity of unsaturated soils. Soil Sci Soc Am J, 44, 892–8, 1980

Vogel T., Cislerova M. On the reliability of unsaturated hydraulic conductivity calculated from the moisture retention curve. Transp Porous Media, 31, 15, doi:10.1007/BF00222683, 1988

Vogel T., van Genuchten M.T., Cislerova M. Effect of the shape of the soil hydraulic functions near saturation on variably-saturated flow predictions. Adv Water Resour, 24, 133–144. doi:10.1016/S0309-1708(00)00037-3, 2001

---

## Author Comment (AC1)

We'd like to thank the reviewer for its careful reading and its useful comments.

The manuscript focuses on the use of a Richards-based solver to reproduce hydrological observations from multiple lysimeters in France. The case study is also used to compare three soil hydraulic models (i.e., Brooks-Corey, van Genuchten-Mualem, a combination of both). The aim is relevant for HESS and somehow interesting, however the manuscript possesses multiple methodological weaknesses:

1. The choice to use the ISBA LSM model, which was conceived to operate on larger scales, to investigate a process at the lysimeter level (and prove a soil physics point: Brooks vs van Genuchten) is questionable. The model solves the Richards equation using a Crank-Nicolson scheme but there are no details about the spatial discretization, boundary conditions, etc. By reading this (https://doi.org/10.1029/2018MS001545), the model seems to use a multi-layer approach based on the finite difference. Widely used vadose zone hydrological model such as HYDRUS or SWAP use schemes that comply with the mass conservative approach proposed by Celia et al. (1990). These models have been widely tested, and would be a more rational choice to investigate processes at the lysimeter level and compare multiple soil hydraulic models.

The ISBA LSM model is applied at global, regional and local scales. Several studies showed the good performance of ISBA at local scale : Boone et al., 2000, Calvet et al., 1999, Decharme et al., 2011. Moreover, studying lysimeters with LSM allows us to verify and to propose improvement directions for simulations at larger scale, such as the integration of a heterogeneous profile with depth.
Moreover, compared to SWAP or HYDRUS models, ISBA solves both the water and energy budgets at a fine temporal scale so as to provide atmospheric models with surface flux (latent and heat fluxes) and boundary conditions (surface temperature and albedo for instance).

The Richards equation is solved numerically using a Crank‑Nicholson implicit time scheme where the flux term is linearized via a one‑order Taylor series expansion and as Hydrus or SWAP, uses a mass conservative method (Decharme et al., 2011, Decharme et al., 2013, Masson et al., 2013). The mixed form of the Richards equation is solved, as it has more robustness with respect to mass balance.

Thank you for noticing that spatial discretization and boundary conditions were missing.
In this study, the soil discretization was adapted to the lysimeter depth and to the measurement depths: 13 soil layers are used, with nodes at 0.01, 0.04, 0.1, 0.2, 0.4, 0.6, 1.0, 1.2, 1.4, 1.6,1.8 and 2 m. A free drainage condition is used at the

bottom of the soil column. Additionally, a one-year simulation  is performed and the output of the simulation are used

We'll add all these informations in the revised version of the article

2.  The whole methodology on the comparison between model predictions and observations is cumbersome to read, not novel, and weak.

Sorry that you didn't enjoy the reading, although reviewer 3 stated that ", the paper is well written and a pleasure to read" . The simulations of the lysimeter data are made using a SVAT model used in regional hydrology, weather and climate models, which is not that common, and include even an interactive vegetation scheme.  The comparison of the simulation with the observation is made using classical statistical scores, but also, focussing on specific events, which again is not that common. Hovmoller diagrams (figures 9, 10, 12) and Taylor diagram (figure 11) are used. But it seems that the comment focuses mainly on the estimation of the soil parameter, according to the following points:

● No error metric is reported to    compare  multiple  soil  hydraulic  models. Besides fitting (which     should  be  quantified), other metrics should be used to compare also the complexity of the models (e.g., at least Akaike Information  Criterion)
● The calibration procedure should compare time series of modeled and observed soil water quantities (e.g., water contents). An objective function or a likelihood (e.g., NSE, Gaussian, etc) should be selected, and a numerical algorithm should be used to perform the model calibration. Further, parameters uncertainty should be assessed to see how informative are data, and whether the choice of a more complex model is justified. Only after having performed a statistically robust analysis, it is possible to try to explain why BC+VG is better and when. As they are, methods don't      support enough the conclusions, and neither represent a novel contribution to the field.

Thank you for these useful comments. Indeed, we did not provide details on the soil parameters calibration procedure, as we mainly focused on the difference between the soil parameters derived from in-situ data to those derived from pedotransfer functions.  We will add the  details in the revised  version of the article.

The calibration of the soil parameters was performed using two methods: i) an objective least squares function which minimizes the sum of the squares of the

deviations and corresponds to maximizing the likelihood with a normal distribution (function nls of rstudio). ii) the package SoilHyP (Dettmann et al., 2022) which uses the Shuffled Complex Evolution (SCE) optimization.

The two methods converge on very closed estimated parameters ( $\delta b: 1.6;\ \delta n: 0.01;\ \delta wsat: 0.0085\ (m^3 m^{-3});\ \delta \alpha: 2\ (m^{-1});\ \delta \Psi sat: 0.019\ (m)$ ).

Statistical scores and errors metrics on the estimation of the soil water retention curves (WRC) are presented in Table R1.1 for the BC66 and VG80 relationships at each depth. On the entire fit and close to saturation (> $\omega_{sat}$ * 0.9), values median, minimal and maximal are presented. The fitted are generally better in depth, and a better r² for VG80 than for BC66 near to saturation. Near saturation, the NRMSE is also better for VG80 than BC66.

Table R1.1 : Statistical scores (regression, r²) and errors metrics (Normalized Root Mean Square Error, NRMSE, and Akaike information criterion, AIC) for the BC66 and VG80 relationships at each depth (20-50-100 and 150 cm) on the total entire fit (Total) and close to saturation (> $\omega_{sat}$ * 0.9) for all lysimeters for soil water retention curves (WRC).

**Total**

|  | r² | | NRMSE | | AIC | |
|---|---|---|---|---|---|---|
|  | BC66 | VG80 | BC66 | VG80 | BC66 | VG80 |
| **20cm** | | | | | | |
| median | 0,891 | 0,886 | 0,337 | 0,34 | 1755,675 | 1777,015 |
| min | 0,846 | 0,84 | 0,229 | 0,189 | 23,154 | 34,243 |
| max | 0,958 | 0,966 | 0,393 | 0,401 | 5124,888 | 4938,271 |
| **50cm** | | | | | | |
| median | 0,69 | 0,84 | 0,67 | 0,4295 | 931,76 | 550,886 |
| min | 0,024 | 0,666 | 0,421 | 0,156 | -1045,18 | -1020,23 |
| max | 0,897 | 0,976 | 1,761 | 0,693 | 5828,86 | 5773,81 |
| **100cm** | | | | | | |
| median | 0,968 | 0,967 | 0,2285 | 0,1975 | -2003,996 | -2061,039 |
| min | 0,691 | 0,684 | 0,111 | 0,095 | -3454,685 | -3501,568 |
| max | 0,989 | 0,991 | 0,624 | 0,563 | 2368,327 | 2270,159 |
| **150cm** | | | | | | |
| median | 0,966 | 0,97 | 0,244 | 0,2435 | -1842,222 | -1546,381 |
| min | 0,711 | 0,66 | 0,122 | 0,159 | -5659,271 | -5651,963 |
| max | 0,991 | 0,987 | 0,608 | 0,7 | 6884,241 | 6852,803 |

**Close to Saturation**

|  | r² | | NRMSE | | AIC | |
|---|---|---|---|---|---|---|
|  | BC66 | VG80 | BC66 | VG80 | BC66 | VG80 |
| **20cm** | | | | | | |
| median | 0,98 | 0,99 | 1,136 | 0,757 | -129,256 | -216,541 |
| min | 0,967 | 0,978 | 1,11 | 0,665 | -249,818 | -419,598 |
| max | 0,983 | 0,995 | 1,212 | 1,7 | 26,347 | 4,815 |
| **50cm** | | | | | | |
| median | 0,938 | 0,926 | 0,629 | 0,483 | 220,637 | -150,892 |
| min | 0,739 | 0,75 | 0,383 | 0,369 | -105,151 | -868,689 |
| max | 0,99 | 0,999 | 0,775 | 1,041 | 2154,151 | 1429,798 |
| **100cm** | | | | | | |
| median | 0,932 | 0,953 | 0,493 | 0,568 | 111,815 | -84,368 |
| min | 0,739 | 0,773 | 0,219 | 0,161 | -118,14 | -226,769 |
| max | 0,979 | 0,987 | 1,12 | 1,412 | 770,041 | 642,032 |
| **150cm** | | | | | | |
| median | 0,849 | 0,931 | 0,403 | 0,313 | 247,332 | 198,666 |
| min | 0,543 | 0,67 | 0,187 | 0,17 | -209,217 | -2331,309 |
| max | 0,964 | 0,996 | 0,598 | 1,02 | 720 | 696,064 |

The calibration of the hydraulic conductivity curves (HCC) was also realized. For each depth, we estimated the hydraulic conductivity at saturation with the water volumetric content and the drainage at 2m for at least 3 years by averaging the drainage values for each water volumetric content value. Statistical scores are shown in Table R1.2 for the four relations. RMSE are better at depth with closer differences than at surface, and always better for VGBC (<0.5 mm/h). AIC are relatively closed between the four relations.

Table R1.2 : Statistical scores (regression, r²) and errors metrics (Root Mean Square Error, RMSE, and Akaike information criterion,AIC) for the BC66 and VG80 relationships at each depth (20-50-100 and 150 cm) for calibration for

hydraulic conductivity curve (HCC).

| | r² | | | | RMSE (mm/h) | | | | AIC | | | |
|---|---|---|---|---|---|---|---|---|---|---|---|---|
| | BC66 | VG80 | VGc | VGBC | BC66 | VG80 | VGc | VGBC | BC66 | VG80 | VGc | VGBC |
| | | | | | | 20cm | | | | | | |
| median | **0,734** | 0,252 | 0,362 | 0,475 | 0,545 | 0,626 | 1,201 | **0,467** | -20,406 | **-230,292** | 92,441 | -35,782 |
| min | 0,318 | 0,214 | 0,153 | 0,242 | 0,476 | 0,546 | 0,604 | 0,427 | -34,891 | -272,773 | -28,618 | -67,644 |
| max | 0,766 | 0,361 | 0,415 | 0,482 | 0,928 | 1,008 | 5,049 | 0,767 | 39,966 | -24,493 | 178,235 | -32,912 |
| | | | | | | 50cm | | | | | | |
| median | **0,601** | 0,295 | 0,415 | 0,332 | 0,669 | 0,504 | 0,758 | **0,426** | **-27,32** | -27,592 | -26,061 | -25,694 |
| min | 0,346 | 0,106 | 0,181 | 0,185 | 0,199 | 0,223 | 0,345 | 0,132 | -37,285 | -62,605 | -36,637 | -46,585 |
| max | 0,846 | 0,524 | 0,712 | 0,653 | 0,82 | 0,788 | 9,66 | 0,708 | 65,976 | -12,016 | 226,709 | -11,994 |
| | | | | | | 100cm | | | | | | |
| median | **0,688** | 0,417 | 0,578 | 0,36 | 0,352 | 0,425 | 0,407 | **0,3** | -23,813 | -25,735 | -23,651 | **-30,998** |
| min | 0,398 | 0,125 | 0,208 | 0,19 | 0,283 | 0,366 | 0,352 | 0,187 | -63,182 | -255,385 | -152,687 | -65,784 |
| max | 0,808 | 0,876 | 0,876 | 0,888 | 0,649 | 0,659 | 0,809 | 2,034 | 51,406 | -16,898 | 65,95 | 142,02 |
| | | | | | | 150cm | | | | | | |
| median | **0,599** | 0,248 | 0,472 | 0,307 | 0,548 | 0,659 | 0,612 | **0,485** | **-34,007** | -27,345 | -30,193 | -32,703 |
| min | 0,194 | 0,132 | 0,155 | 0,184 | 0,298 | 0,366 | 0,357 | 0,175 | -59,104 | -468,014 | -32,471 | -88,413 |
| max | 0,96 | 0,584 | 0,801 | 0,667 | 0,747 | 0,742 | 0,946 | 0,675 | 59,053 | -17,797 | 81,5 | -21,512 |

Specific comments:

L15-20 Not really. Drainage is the amount of water that bypasses the root zone.

It is true that drainage can be associated with different fluxes, which is why we provided the definition. Indeed, if drainage is sometimes associated to the amount of water that bypasses the root zone (Silburn et al., 2013) it is also traditionally associated to the part of precipitation that flows through the first meters of soil down to the aquifer (Philip et al., 1969, Whisler et al., 1970). We will add these references to the revised version of the article.

L47-50 Nonlinearity cannot be a source of criticism, otherwise an endless number of equations used in environmental modeling should be "criticized". I would remove this part. Richards equation is not perfect, but we are still far from finding a viable, widely used, and extensively validated alternative.

We agree with your comments, and we agree to remove this part.

L56 BC66 has that sharp singular point near the air-entry pressure that makes it not very stable. (https://doi.org/10.1029/93WR03238). Authors indeed discuss this point later. However, more specific references are needed to prove your point that BC66 is more numerically stable than VG80.

Thank you for highlighting this point. The non-linear form of the VG80 hydraulic conductivity has a high reduction at pressures near saturation in particular for n values close to 1, and has numerical convergence problems (Van Genuchten 1980, Vogel 2000), notably when parameters n and m are dependent (Dourado et al., 2011).

We proposed to modify the article like this : The non-linear form of the VG80 hydraulic conductivity has a high reduction at pressures near saturation in particular for n values close to 1, and has numerical convergence problems (Van Genuchten 1980, Vogel 2000), notably when parameters n and m are dependent (Dourado et al., 2011).

Data: Please add details about TDR sensors (e.g., type, accuracy, calibration type) and tipping bucket resolution

On the GISFI site, TDR probes are RIME-PICO32 sensors with internal TDR-electronics. They are set horizontally and record the water content in cm3 cm− 3 ( ± 0.01) on an hourly basis. The calibration was performed on two measurements, one in dry and one in water-saturated condition. In the OPE site, soil moisture sensors used (UMP-1Umwelt Geräte Technik GmbH) are based on frequency domain reflectometry (FDR) method and measure local change in dielectric permittivity.

Tipping bucket resolution is 0.1mm.h-1 on the two sites.

L124-125 Is the heat transport included in the numerical simulation of lysimeters? If yes, key equations should be provided. Otherwise, it should be removed from the text.

Yes, as stated, the heat transport is solved. Most Land Surface Models use multilayer soil diffusion schemes, which solves mass and heat diffusive equations.  We chose  not to provide the equations, since only a global statics is provided on the ability of ISBA, to solve soil temperature line 229.
We propose to add heat transport and temperature equations in the appendix. The surface soil temperature evolves to the surface heat flux rate G (W m²) for N soil layers by the use of the classical one‑dimensional Fourier law (Boone et al., 2000, Decharme et al., 2013).

$$\frac{\delta Ts}{\delta t} = C_T [ G - \frac{\overline{\lambda 1}}{\overline{\Delta z1}} * (T_1 - T_2 )]$$

$$\frac{\delta Ti}{\delta t} = \frac{1}{C_{gi}} \frac{1}{\Delta zi} [ \frac{\overline{\lambda i-1}}{\overline{\Delta zi-1}} * (T_{i-1} - T_i ) - \frac{\overline{\lambda i}}{\overline{\Delta zi}} * (T_i - T_{i+1} )]$$

where $\Delta zi$ (m) is the thickness of the layer i,  $\overline{\Delta z1}$  (m) is the thickness between two consecutive layer nodes, $C_{gi}$ (J m−3 K−1) is the total soil heat capacity, and $\lambda i$ (W m−1 K−1) is the inverse‑weighted arithmetic mean of the soil thermal conductivity at the interface between two consecutive nodes.

L130-140 This part should be moved after the Richards equation, and should describe how it is connected to the sink term S(z). Key equations should provided. Citing refs is good, but the manuscript should stand by itself.

Thank you for your judicious suggestion, we will move this part.

The sink term S(z) is the evapotranspiration from vegetation and evaporation from the bare soil.

L147-148 what is the discretization of the soil profile? What are the boundary conditions used?

Infiltration in ISBA is computed at a 5-minute time steps as the precipitation that drops through the canopy and reaches the first layer of soil. The Green-Ampt approach is used to determine the maximum amount of water that infiltrates the soil.

For the boundary conditions, please refers to the answer to your question #1 above

Figure1. Very confusing. It is difficult to appreciate differences. What are the dashed lines? Figure+Caption should be self-explanatory

We are sorry that the figure seems confusing and we will improve the caption. The figure has two main goals: i) to illustrate the variability of the soil properties between two soil columns and within the soil profile and ii) to illustrate the difference between BC66 and VG80.

We agree that the 1st point is easier to see than the 2nd one, and this is why the dashed lines were added. The dashed lines are the derived values of the water content at saturation and matric potential at saturation for BC66 which is also the alpha parameter for VG80. Therefore, the vertical lines help to show the differences between the expressions of BC66 and VG80, as the value of the matrix pressure cannot have lower values for BC66 (in absolute value), while it can with VG80.

This will be added in the caption, and added line 180.

L176 There is not a single error metric to support the conclusion that one formulation is better than the other. It is really puzzling to see that.

Thank you for raising this point. Indeed, it appears quite clearly from such a graph that the VG80 expression of matrix potential is closer to the observation

close to the saturation than BC66, and the article mainly focussed on how this impacts the simulation of the soil water content and soil water drainage, with numerous statistics on these comparisons.

But of course, we can add the statistics on the soil parameters calibration. . Please, refers to the answer to your question  #2 above  that provides the full statistics.

L192 VG80 not stable for n<1.3?! Never experienced something like this. Indeed, I agree with the Authors that n>1.1 is a good constraint.

Thank you for sharing  your agreement. The relation m=1-1/n; applicated in this study for the VG80 relations, is not recommended for n <1.25 and n > 6  (Van Genuchten 1985). The non-linear form of the VG80 hydraulic conductivity has a high reduction at pressures near saturation in particular for n values close to 1, and has numerical convergence problems (Van Genuchten 1980, Vogel 2000).

L197 Having a highly negative tortuosity is not recommended. Actually Schaap suggests a value of -1 (https://doi.org/10.2136/sssaj2000.643843x)

Thank you for raising  this point. The NRMSE score on the comparison between observed and simulated drainage of the sensitivity tests are presented in Table R1.3. Best results were obtained for a tortuosity fixed at 0.5 for OPE lysimeters (O1-O2 and O4); and -5 for GISFI lysimeters (G1-G2-G3-G4).

Table R1.3 : Normalized Root Mean Square Error (NRMSE) scores from simulations with VG80 for each lysimeters, with variation of the parameter l.

| | NRMSE | | | | | | |
|------|-------|-------|-------|-------|-------|-------|-------|
| **l** | **G1** | **G2** | **G3** | **G4** | **O1** | **O2** | **O4** |
| -5 | 0,738 | 0,851 | 0,861 | 0,81 | 0,867 | 0,929 | 0,777 |
| -2 | 0,741 | 0,875 | 0,989 | 0,813 | 0,861 | 0,946 | 0,753 |
| -1 | 0,763 | 0,93 | 0,968 | 0,81 | 0,873 | 0,931 | 0,763 |
| -0,5 | 0,738 | 0,898 | 0,969 | 0,855 | 0,873 | 0,926 | 0,763 |
| 0,5 | 0,774 | 1,038 | 0,876 | 0,817 | 0,694 | 0,817 | 0,708 |
| 1 | 0,929 | 1,185 | 0,899 | 0,826 | 0,812 | 0,826 | 0,715 |
| 2 | 0,878 | 1,144 | 0,861 | 0,849 | 0,694 | 0,824 | 0,708 |
| 5 | 0,966 | 1,031 | 0,948 | 0,869 | 0,77 | 0,83 | 0,736 |

References

Boone, A., Masson, V., Meyers, T., & Noilhan, J. (2000). The Influence of the Inclusion of Soil Freezing on Simulations by a Soil–Vegetation–Atmosphere Transfer
Scheme. JOURNAL OF APPLIED METEOROLOGY , 39 , 26.

Decharme, B., Boone, A., Delire, C., & Noilhan, J. (2011). Local evaluation of the
Interaction between Soil Biosphere Atmosphere soil multilayer diffusion scheme
using four pedotransfer functions. Journal of Geophysical Research, 116 (D20)D20126. doi: 10.1029/2011JD016002

Decharme, B., Martin, E., and Faroux, S.: Reconciling soil thermal and hydrological lower boundary conditions in land surface models, Journal of Geophysical Research: Atmospheres, 118, 7819–7834, 2013

Calvet, J.-C., Bessemoulin, P., Noilhan, J., Berne, C., Braud, I., Courault, D., Fritz, N., Gonzalez-Sosa, E., Goutorbe, J.-P., Haverkamp, R.,Jaubert, G., Kergoat, L., Lachaud, G., Laurent, J.-P., Mordelet, P., Olioso, A., Péris, P., Roujean, J.-L., Thony, J.-L., Tosca, C., Vauclin, M., and Vignes, D.: MUREX: a land-surface field experiment to study the annual cycle of the energy and water budgets, Annales Geophysicae,470 17, 838–854, https://doi.org/10.1007/s00585-999-0838-2, 1999.

Dettmann, U., Andrews, F., and Donckels, B.: Package 'SoilHyP', 2022.

Dourado Neto, D., de Jong van Lier, Q., van Genuchten, M., Reichardt, K., Metselaar, K., and Nielsen, D.: Alternative Analytical Expressions for the General van Genuchten–Mualem and van Genuchten–Burdine Hydraulic Conductivity Models, Vadose Zone Journal, 10, 618–623, https://doi.org/https://doi.org/10.2136/vzj2009.0191, 2011.

Masson, V., Le Moigne, P., Martin, E., Faroux, S., Alias, A., Alkama, R., Belamari, S., Barbu, A., Boone, A., Bouyssel, F., et al.: The SURFEXv7. 2 land and ocean surface platform for coupled or offline simulation of earth surface variables and fluxes, Geoscientific Model Development, 6, 929–960, 2013

PHILIP, J.: Theory of Infiltration, vol. 5 of Advances in Hydroscience, pp. 215–296, Elsevier, https://doi.org/https://doi.org/10.1016/B978-1-605
4831-9936-8.50010-6, 1969

Silburn, D., Montgomery, J., McGarry, D., Gunawardena, T., Foley, J., Ringrose-Voase, A., and Nadelko, A.: Deep drainage under irrigated cotton in Australia: a review, pp. 40–58, 2013.

van Genuchten, M. T.: A Closed-form Equation for Predicting the Hydraulic Conductivity of Unsaturated Soils, Soil Science Society of America Journal, 44, 892–898, https://doi.org/10.2136/sssaj1980.03615995004400050002x, 1980.

Van Genuchten, M. and Nielsen, D.: On Describing and Predicting the Hydraulic Properties of Unsaturated Soils, Annales Geophysicae, 3,615–628, 1985.

Vogel, T., van Genuchten, M., and Cislerova, M.: Effect of the shape of the soil hydraulic functions near saturation on variably-saturated flow predictions, Advances in Water Resources, 24, 133–144, https://doi.org/10.1016/S0309-1708(00)00037-3, 2000.

Whisler, F. D. and Bouwer, H.: Comparison of methods for calculating vertical drainage and infiltration for soils, Journal of Hydrology, 10,1–19, 1970.

---

## Author Comment (AC2)

This manuscript presents the results of an LSM application using two different approaches according to BC and VG to reproduce soil water mass, volumetric water content, and drainage water flux volume observed in seven lysimeters over a period of more than five years. Furthermore, approaches by Braud et al. (1995) and Valiantzas (2011) are tested to simulate the soil hydrology of these lysimeters. They derived hydrodynamic parameters directly from the observation and compare them with several pedotransfer functions commonly used by LSMs. The LSM used has a multilayer diffusion approach of the Interaction-Soil-Biosphere-Atmosphere (ISBA) model, which solves a variant of the Richards equation.

- The term "drainage" in this context means the transit of a liquid through a porous medium. In the present case, it is water through the upper soil layers. It is neither a quantity nor a volume. Therefore, if the amount of water is to be addressed this must be explicitly stated as drainage water.

Thank you for this comment. We agree with you and have agreed to change the term "drainage" by "drainage water".

- For the lysimeters, the experimental setup is sufficiently described, but the lower boundary condition is not mentioned in detail as a special feature of the lysimeter. Since drainage in particular is considered as a special aspect, this has to be described in detail for the lysimeters. Especially the consequences/impacts of the chosen design on the drainage amount of water must be discussed. Otherwise, it is assumed here that the lower boundary layer of the lysimeter corresponds to a naturally layered soil, and this is de facto not the case.

Yes, some information on the setting of the simulation was missing.

In this study, the soil discretization was adapted to the lysimeter depth and to the measurement depths: 13 soil layers are used, with nodes at 0.01, 0.04, 0.1, 0.2, 0.4, 0.6, 1.0, 1.2, 1.4, 1.6,1.8 and 2 m. A free drainage condition is used at the bottom of the soil column.

Such conditions imply that the bottom of the soil should be saturated to generate. You'll see below that this can be observed, at the answer to your question on Line 181.

Additionally, a one-year simulation is performed and the output of the simulation are used

- The methods section on the comparison between the model predictions and the lysimeter observations is very unclearly written and needs a more comprehensible description.

Thank you for these useful comments. Indeed, we did not provide details on the soil parameters calibration procedure, as we mainly focused on the difference between the soil parameters derived from in-situ data to those derived from pedotransfer functions. We will add the details in the revised version of the article.

The calibration of the soil parameters was performed using two methods: i) an objective least squares function which minimizes the sum of the squares of the deviations and corresponds to maximizing the likelihood with a normal distribution (function nls of rstudio). ii) the package SoilHyP (Dettmann et al., 2022) which uses the Shuffled Complex Evolution (SCE) optimization.

The two methods converge on very closed estimated parameters ( $\delta b$: 1.6; $\delta n$: 0.01; $\delta wsat$: 0.0085 $(m^3 m^{-3})$; $\delta\alpha$: 2 $(m^{-1})$; $\delta\Psi sat$: 0.019 $(m)$ ).

Statistical scores and errors metrics on the estimation of the soil water retention curves (WRC) are presented in Table R2.1 for the BC66 and VG80 relationships at each depth. On the entire fit and close to saturation (> $\omega_{sat}$ * 0.9), values median, minimal and maximal are presented. The fitted are generally better in depth, and a better r² for VG80 than for BC66 near to saturation. Near saturation, the NRMSE is also better for VG80 than BC66.

Table R2.1 : Statistical scores (regression, r²) and errors metrics (Normalized Root Mean Square Error, NRMSE, and Akaike information criterion, AIC) for the BC66 and VG80 relationships at each depth (20-50-100 and 150 cm) on the total entire fit (Total) and close to saturation (> $\omega_{sat}$ * 0.9) for all lysimeters for soil water retention curves (WRC).

| | Total | | | | | | | Close to Saturation | | | | | |
| | r² | | NRMSE | | AIC | | | r² | | NRMSE | | AIC | |
| | BC66 | VG80 | BC66 | VG80 | BC66 | VG80 | | BC66 | VG80 | BC66 | VG80 | BC66 | VG80 |
| **20cm** | | | | | | | **20cm** | | | | | | |
| median | 0,891 | 0,886 | 0,337 | 0,34 | 1755,675 | 1777,015 | median | 0,98 | 0,99 | 1,136 | 0,757 | -129,256 | -216,541 |
| min | 0,846 | 0,84 | 0,229 | 0,189 | 23,154 | 34,243 | min | 0,967 | 0,978 | 1,11 | 0,665 | -249,818 | -419,598 |
| max | 0,958 | 0,966 | 0,393 | 0,401 | 5124,888 | 4938,271 | max | 0,983 | 0,995 | 1,212 | 1,7 | 26,347 | 4,815 |
| **50cm** | | | | | | | **50cm** | | | | | | |
| median | 0,69 | 0,84 | 0,67 | 0,4295 | 931,76 | 550,886 | median | 0,938 | 0,926 | 0,629 | 0,483 | 220,637 | -150,892 |
| min | 0,024 | 0,666 | 0,421 | 0,156 | -1045,18 | -1020,23 | min | 0,739 | 0,75 | 0,383 | 0,369 | -105,151 | -868,689 |
| max | 0,897 | 0,976 | 1,761 | 0,693 | 5828,86 | 5773,81 | max | 0,99 | 0,999 | 0,775 | 1,041 | 2154,151 | 1429,798 |
| **100cm** | | | | | | | **100cm** | | | | | | |
| median | 0,968 | 0,967 | 0,2285 | 0,1975 | -2003,996 | -2061,039 | median | 0,932 | 0,953 | 0,493 | 0,568 | 111,815 | -84,368 |
| min | 0,691 | 0,684 | 0,111 | 0,095 | -3454,685 | -3501,568 | min | 0,739 | 0,773 | 0,219 | 0,161 | -118,14 | -226,769 |
| max | 0,989 | 0,991 | 0,624 | 0,563 | 2368,327 | 2270,159 | max | 0,979 | 0,987 | 1,12 | 1,412 | 770,041 | 642,032 |
| **150cm** | | | | | | | **150cm** | | | | | | |
| median | 0,966 | 0,97 | 0,244 | 0,2435 | -1842,222 | -1546,381 | median | 0,849 | 0,931 | 0,403 | 0,313 | 247,332 | 198,666 |
| min | 0,711 | 0,66 | 0,122 | 0,159 | -5659,271 | -5651,963 | min | 0,543 | 0,67 | 0,187 | 0,17 | -209,217 | -2331,309 |
| max | 0,991 | 0,987 | 0,608 | 0,7 | 6884,241 | 6852,803 | max | 0,964 | 0,996 | 0,598 | 1,02 | 720 | 696,064 |

The calibration of the hydraulic conductivity curves (HCC) was also realized. For each depth, we estimated the hydraulic conductivity at saturation with the water volumetric content and the drainage at 2m for at least 3 years by averaging the drainage values for each water volumetric content value. Statistical scores are shown in Table R2.2 for the four relations. RMSE are better at depth with closer differences than at surface, and always better for VGBC (<0.5 mm/h). AIC are relatively closed between the four relations.

Table R2.2 : Statistical scores (regression, r²) and errors metrics (Root Mean Square Error, RMSE, and Akaike information criterion,AIC) for the BC66 and VG80 relationships at each depth (20-50-100 and 150 cm) for calibration for hydraulic conductivity curve (HCC).

| | r² | | | | RMSE (mm/h) | | | | AIC | | | |
|---|---|---|---|---|---|---|---|---|---|---|---|---|
| | BC66 | VG80 | VGc | VGBC | BC66 | VG80 | VGc | VGBC | BC66 | VG80 | VGc | VGBC |
| **20cm** | | | | | | | | | | | | |
| median | **0,734** | 0,252 | 0,362 | 0,475 | 0,545 | 0,626 | 1,201 | **0,467** | -20,406 | **-230,292** | 92,441 | -35,782 |
| min | 0,318 | 0,214 | 0,153 | 0,242 | 0,476 | 0,546 | 0,604 | 0,427 | -34,891 | -272,773 | -28,618 | -67,644 |
| max | 0,766 | 0,361 | 0,415 | 0,482 | 0,928 | 1,008 | 5,049 | 0,767 | 39,966 | -24,493 | 178,235 | -32,912 |
| **50cm** | | | | | | | | | | | | |
| median | **0,601** | 0,295 | 0,415 | 0,332 | 0,669 | 0,504 | 0,758 | **0,426** | **-27,32** | -27,592 | -26,061 | -25,694 |
| min | 0,346 | 0,106 | 0,181 | 0,185 | 0,199 | 0,223 | 0,345 | 0,132 | -37,285 | -62,605 | -36,637 | -46,585 |
| max | 0,846 | 0,524 | 0,712 | 0,653 | 0,82 | 0,788 | 9,66 | 0,708 | 65,976 | -12,016 | 226,709 | -11,994 |
| **100cm** | | | | | | | | | | | | |
| median | **0,688** | 0,417 | 0,578 | 0,36 | 0,352 | 0,425 | 0,407 | **0,3** | -23,813 | -25,735 | -23,651 | **-30,998** |
| min | 0,398 | 0,125 | 0,208 | 0,19 | 0,283 | 0,366 | 0,352 | 0,187 | -63,182 | -255,385 | -152,687 | -65,784 |
| max | 0,808 | 0,876 | 0,876 | 0,888 | 0,649 | 0,659 | 0,809 | 2,034 | 51,406 | -16,898 | 65,95 | 142,02 |
| **150cm** | | | | | | | | | | | | |
| median | **0,599** | 0,248 | 0,472 | 0,307 | 0,548 | 0,659 | 0,612 | **0,485** | **-34,007** | -27,345 | -30,193 | -32,703 |
| min | 0,194 | 0,132 | 0,155 | 0,184 | 0,298 | 0,366 | 0,357 | 0,175 | -59,104 | -468,014 | -32,471 | -88,413 |
| max | 0,96 | 0,584 | 0,801 | 0,667 | 0,747 | 0,742 | 0,946 | 0,675 | 59,053 | -17,797 | 81,5 | -21,512 |

- Why these applied models were selected is not convincingly presented, especially since there are more current modeling approaches that promise better simulation of processes and results.

The model ISBA-SURFEX solves both the water and energy budgets at a fine temporal scale so as to provide atmospheric models with surface flux (latent and heat fluxes) and boundary conditions (surface temperature and albedo for instance). It is used in regional hydrological models, as well as in weather forecast and climate models. Therefore, it is important to assess its simulations of the soil water flux.

Like many soil–vegetation– atmosphere–transfer schemes (SVAT), it uses the relation of BC66. Compared to data from lysimeters, it is obvious that this relationship is not the most correct to reproduce WRCs close to saturation. This is why we decided to use the relations of VG80, but the hydraulic conductivity curves of VG80 requires new parameters, and VG80 is unstable close to small values of n.

The BCVG relation is one solution to solve this numerical problem; an approach has been used in other studies (Braud et al., 1995, Valiantzas, 2011), which consists of combining VG80 water retention curves and BC66 hydraulic conductivity curves. We are well aware that BCVG seems not very physical from a theoretical point of view, but from a pragmatic and statistical point of view, water retention curves and HCC can reproduce the observations on lysimeters very well. The advantage of this approach is also the limited number of parameters and can be spatialized easily at regional or global scale in the ISBA surface model. Following the comments of the 3st reviewer, Wolfgand Durner, we also tested an intermediate approach by truncating the pore-size distribution (Iden et al., 2015), which provide promising results (see answer to reviewer 3)

- Lysimeters provide "point" information compared to LSM. Here, indications are missing how this discrepancy is addressed or how lysimeter results could be scaled.

The ISBA LSM model is applied at global, regional and local scales. Several studies showed the good performance of ISBA at local scale : Boone et al., 2000, Calvet et al., 1999, Decharme et al., 2011. Moreover, studying lysimeters with LSM allows us to verify and to propose improvement directions for simulations at larger scale, such as the integration of a heterogeneous profile with depth.

Line 107: A specification of the measurement resolution is missing here.

On the GISFI site, TDR probes are RIME-PICO32 sensors with internal TDR-electronics. They are set horizontally and record the water content in $cm^3$ $cm^{-3}$ ( ± 0.01) on an hourly basis. The calibration was performed on two measurements, one in dry and one in water-saturated condition. In the OPE site, soil moisture sensors used (UMP-1Umwelt Geräte Technik GmbH) are based on frequency domain reflectometry (FDR) method and measure local change in dielectric permittivity.

Tipping bucket resolution is 0.1mm.h-1 on the two sites.

Line 181 ff: Is this the case? It is often stated that in zero-tension lysimeters, the seepage water formation takes place under water-saturated conditions. I am not aware of any study that has decisively investigated this. Of course, small-scale saturated structures are also conceivable with corresponding fingering. Is this the case? Especially before the background of a very heterogeneous material of a former industrial site, which was filled manually into lysimeters. Hydrophobic structures are also conceivable.

Yes it is the case. In the Figure R2.1, variation of the total hydraulic pressure head between 50 and 150 cm ($\frac{dH}{dz}$) of depth is represented for each lysimeter. Lines blue and red represent a gradient equal to 1 and 0 respectively. It can be observed that the gradient is in most cases equal or very close to 1. Some variations exist, in particular for lysimeter G3, because of strong and deep roots.

Nevertheless, when the saturation of volumetric content in the soil is reached, the gradient is equal to 1, and we can assume at this moment, ksat is equal to observed drainage. The lower boundary condition is a free drainage.

The process of hydrophobicity is negligible in these lysimeters.

.

[Figure]

Figure R2.1 : Variation of the total hydraulic pressure head between 50 and 150 cm ($\frac{dH}{dz}$) of depth is represented for each lysimeter. Lines blue and red represent a gradient equal to 1 and 0 respectively

Line 200 ff: Should be discussed later, because it is manual filling with disturbed profiles. This has an impact on the parameter estimation.

We propose to add in the discussion this point :

Our observations show that the soil hydrodynamic parameters in each lysimeter are strongly heterogeneous with depth, while LSMs generally use homogeneous profiles. Lysimeters were filled with preserved soil columns at the OPE, and manually at the GISFI. The fact that the lysimeters are filled in a manual way can explain this evolution, although they have been filled to preserve the bulk density of the soil. Factors like compaction and structuration can be present on lysimeters (Durner, 1994, Séré et al., 2012).

 Using additional experiments with such homogeneous profiles, we found that even if the simulated soil water and drainage dynamics remain acceptable compared to the observations, all the skill scores are worsen compared to the experiments with a heterogeneous profile.

Line 233 ff: "At the bottom of the soil" What do you mean by this? This wording is very unclear or does not make sense.

We meant that the drainage is measured at 2 m of depth, the depth of the lysimeters. We'll make it clearer in the revised version.

Line 242: Masses of what? Water?

Yes, it is well the total water mass on lysimeter.

We proposed to modify the article like this : 4.1.1 "Total Water Mass".

Line 266f: I do not understand this argumentation. The temporal resolution is criticized as limiting and therefore I reduce the temporal resolution even more or aggregate the data?

We choose to aggregate the data on daily drainage to not be limited by the threshold resolution of the measurements, since at the hourly time steps, the measurements are most often 0 or 0.1 mm…. Aggregating at a daily time step provides observations that are easier to interpret .

Line 265ff: In order to be able to classify the different seepage water quantities, a distinction must be made between vegetated and unvegetated lysimeters. This has been done. But to be able to investigate or classify the differences between the vegetated lysimeters, measurements of the crop development (LAI) or the crop yield (harvest amount), etc. are absolutely needed. Only with this information different ETp results can be classified.

We have no information of the crop yield for these lysimeters, notably because vegetation were not harvested (except for G4). No LAI measurements were available on these lysimeters, but LAI is simulated (line 220-225). For more precision, we propose to add a figure with a mean annual cycle of the LAI for lysimeters with vegetation and the daily simulated LAI in the supplementary data (Figures R2.2 and R2.3.

Moreover, the water budget of each lysimeter is also estimated (Figure 8 in the paper) and can give an information of the ETp.

.

[Figure]

Figure R2.2 : Mean LAI cycle simulated for lysimeters for each lysimeter

[Figure]

Figure R2.3 : Daily evolution of the LAI simulated for each vegetated lysimeter

Line 289 ff: Also, non rainfall water like dew, hoar frost, etc.

Thank you for this comment. We specified this water budget is conducted on an annual scale and dew/rime is negligible at this scale, although dew deposition is simulated by the model, and is considered in the balance as negative evaporation. Also, there was no freezing/irrigation on these sites during the measurements.

Table 1: Regarding the contents of the table: am I correct in assuming that the remainder is 100% silt? If not, what then? But should still be presented in more detail for clarity. To standardize the presentation, the number of decimal places should be the same for all data.

Yes, you are right. Table 1 indicates the proportions of clay and sand. For example, G4 lysimeter at 20 cm, is composed of 32% clay and 25% of sand, the remainder is the proportion of silt (43% here). We will modify this table by adding

the proportion in silt as well as putting the same number of decimal for more comprehension.

Table R2.3

**Table 1.** Description of lysimeters: filling method, soil type, vegetation cover, number of texture observations, and textures (in % of clay, sand ans silt) at different depths. HyCa stands for Hypereutric Cambisol

| Site | GISFI experimental station | | | | OPE experimental station | | |
|---|---|---|---|---|---|---|---|
| Lysimeters | G1 | G2 | G3 | G4 | O1 | O2 | O3 |
| | Fill | Fill | Fill | Monolith | Monolith | Monolith | Monolith |
| Soil | Technosol | Technosol | Technosol | Cambisol | HyCa | Cambisol | HyCa |
| Soil cover | bare soil | bare soil | Alfalfa | Grass | Grass | Grass | Grass |
| Layers | 1 | 1 | 1 | 4 | 6 | 1 | 6 |
| Bulk density (kg.m$^{-3}$) | 1300 | 1300 | 1300 | 1300 | 1700 | 1700 | 1700 |
| USDA soil type | Sandy Clay Loam | Sandy Clay Loam | Sandy Clay Loam | Clay Loam | Clay | Clay Loam | Silty Clay Loam |
| (Sand,Clay,Silt) (%,%,%) | | | | | | | |
| Homogeneous | (61.6, 14.3, 24.1) | (61.6, 14.3, 24.1) | (62.4, 15.2, 22.4) | (32.0, 25.0, 43.0) | (3.0, 36.0, 61.0) | (31.0, 41.0, 28.0) | (18.0, 47.0, 36.0) |
| 0.2m | "" | "" | "" | (20.0, 15.0, 75.0) | (11.0, 4.0, 85.0) | (50.4, 18.0, 31.6) | (24.0, 28.0, 48.0) |
| 0.5m | "" | "" | "" | (17.0, 26.0, 57.0) | (0.0, 67.0, 37.0) | (42.0, 27.0, 31.0) | (16.0, 53.0, 31.0) |
| 1m | "" | "" | "" | (34.0, 33.0, 33.0) | (0.0, 19.0, 81.0) | (22.0, 6.0, 72.0) | (16.0, 53.0, 31.0) |
| 1.5m | "" | "" | "" | (56.0, 24.0, 20.0) | (0.0, 19.0, 81.0) | (22.0, 6.0, 72.0) | (16.0, 53.0, 31.0) |

Figure 3, 5-7: Measurement units of the Y-axes are missing

The units were provided in the caption but not in the figures. We will add the units in the plots for a better understanding.

Furthermore, I share the remarks of the reviewer 1.

Durner, W.: Hydraulic conductivity estimation for soils with heterogeneous pore structure, Water resources research, 30, 211–223, 1994.

Séré, G., Ouvrard, S., Schwartz, C., Pey, B., and Morel, J.-L.: Identification of hydric functioning patterns during the early pedogenesis of a Constructed Technosol, in: 1. International Conference and Exploratory Workshop on Soil Architecture and Physico-chemical Functions "CESAR", pp. 426–p, Aalborg University; Faculty of Agricultural Sciences, Aarhus University, 2010.

---

## Author Comment (AC3)

We would like to thank Wolfgang Durner for its very useful review.

Especially, thank you for pointing out the improved version of the Van Genuchten proposed by Iden et al., 2015 which we were not aware of.

We integrated these new closing equations and run new simulations. The results are very good, and we will incorporate them into a new version of the article.

As we mentioned in our article, although Van Genuchten (1980) proposed an improvement in the closure equation compared to Brooks & Corey (1966) , the meteorological and climate modeling community that uses Land Surface Models (LSMs) usually use the Brooks & Corey closure equations, as they are more numerically stable and necessitate less parameters. The introduction of a proposed Iden et al. (2105) air-entry suction allows a major advance for LSM modeling.

This progress is made possible by your involvement in this peer review process and we are very grateful

In the following we provide details answers to your questions and comments.

The goal of this study is to use a LSM to reproduce soil water mass, volumetric water content and drainage flux observed in seven lysimeters during a period of more than five years. The simulations are performed with three different parametrizations for the soil hydraulic properties, namely the common functions of (i) Brooks and Corey (1966) [BC66], (ii) van Genuchten-Mualem (1980) [VG80], and additionally (iii) a previously proposed "hybrid" approach, where the VG80 retention function is combined with the BC66 conductivity function [VGBC]. Considering standard diagnostic variables, the authors find a best performance with the VGBC approach and a worst performance for the VG80 approach. Ancillary studies further investigated the replacement of heterogeneous, layered soils with uniform properties. The authors note that all the results deteriorate compared to the tests with a heterogeneous profile and conclude that the vertical heterogeneity of the soil hydrodynamic parameters must be taken into account. Finally, the use of PTF-derived hydraulic properties in the simulation is compared to the use of parameters based on in-situ measurements of water content and matrix potential. Basically, the in-situ derived VG80 parameters n differ strongly from the PTF-derived values, leading to significantly different simulation results, which are worse for the PTF-based simulations.

**General Comment**

Using lysimeter data in soil hydrologic analyses is always exciting because it shows us what we do and do not understand about soil hydrologic processes.

And it gives us clues about the reliability or unreliability of local soil moisture sensor readings. So this work is a very good exercise, and the use of lysimeter data for this type of study has merit. Furthermore, the paper is well written and a pleasure to read. However, I have a problem with the main idea of the paper, which is to propose the use of a VGBC parameterization in soil hydrology.

The approach taken by the authors here may be nice from a numerical point of view, but from a physical point of view it must be considered unsuitable. The reason, in short, is that a smooth, continuously derivable soil water retention function (WRC) is combined with an incompatible shaped hydraulic conductivity function (HCC). This does not make physical sense. Furthermore, the comparison of the three functional approaches is most likely biased by problems with the VG80 model when it is used to parameterize the WRC of a fine-grained soil or a soil with a broad pore size distribution (indicated by a parameter value of $n$ close to 1). In such cases, the VG80 hydraulic conductivity curve (HCC) exhibits an abrupt drop near saturation (Durner, 1994). This drop (i) leads to a severe underestimation of the hydraulic conductivity when the prediction of HCC is based on the measured saturated hydraulic conductivity (Vogel et al., 2001) and (ii) negatively affects the performance of the numerical solvers, i.e., their stability and accuracy during the transition between saturated and unsaturated conditions (Ippisch et al., 2006). This could be the reason why the authors of this study limit the smallest value of n to a threshold value (arbitrarily chosen for numerical reasons) of n > 1.09 and also find in the performance comparison that the VG80 model leads to overly wet soil profiles. To eliminate the well-known artifact of the VG80 conductivity curve for soils with small $n$ three main approaches have been proposed in the past. They are (i) shifting the entire pore-size distribution by an air-entry value (Kosugi, 1994), (ii) introducing an explicit air-entry pressure into the van Genuchten model (Vogel and Cislerova, 1988; Vogel et al., 2001; Ippisch et al., 2006), or (iii) truncating the pore-size distribution (Malama and Kuhlmann, 2015; Iden et al., 2015). Some other workarounds have also been proposed (e.g.Schap and van Genuchten, 2006). The second approach is historically the most commonly used. It effectively eliminates the drop in HCC near saturation and the associated negative effects on the behavior of the numerical solvers of the Richards equation. The drawback, however, is that the resulting WRC is no longer continuously differentiable, i.e., the soil water capacity function becomes discontinuous, which in turn can cause problems with numerical solvers of the Richards equation in some situations. The VGBC approach adopted by the authors in their study is basically a variant of approach iii, i.e., they leave the RETC in its original smooth form, but limit the conductivity to a constant maximum value reached at an "air entry" value (somehow arbitrarily constructed for soils with small n values).

It is certainly fair to try such an approach. However, the paper fails to convince me that this approach is worth pursuing. An important reason for this opinion is the fact that the results in the paper are not fully comprehensible to the reader, since crucial information about the modeling scenario and especially the hydraulic properties of the soil is missing. So we have to believe the results or not. In addition, the paper has shortcomings, especially in the methods section of soil hydrology (Section 3.1), as will be presented later in this review. As is so often the case in soil hydrology, the devil is in the details, and therefore I recommend reviewing the study again in detail, taking into account the comments below. Specifically, I recommend repeating the analysis (or adding a scenario to the analysis) with modified BC80 HCC functions the remove the above mentioned VG80 artifact. This is probably most straightforward using the HCC functions of Iden et al.(2015, Eq. 18 therein). If desired, I [Wolfgang Durner] would be willing to provide the authors with the respective HCC functions for their lysimeter's retention curves.

**SPECIFIC COMMENTS**

1. Basic approach:

As mentioned before, I believe that a mixture of VG RETC and BC HCC is not a good solution to solve the problems with VG80, even though others have tried it and it leads to smooth simulations. The reason is that this is simply unphysical. Above the air entry value, we have significantly increasing water contents, but constant conductivity. This contradicts the soil hydrological evidence.

The model ISBA-SURFEX uses, like many soil–vegetation–atmosphere–transport schemes (SVAT) models, the relation of BC66. Compared to data from lysimeters, it is obvious that this relationship is not the most correct to reproduce WRCs close to saturation. This is why we decided to use the relations of VG80, but the HCC relation of VG80 requires new parameters, and is unstable close to small values of n.

The BCVG relation is one solution to solve this numerical problem; This approach was used in previous studies (Braud et al., 1995, Valiantzas, 2011). It,consists in combining VG80 WRC and BC66 HCC. We are well aware that BCVG seems not very physical from a theoretical point of view, but it is realistic from a statistical and pragmatic point of view, SWC and HCC can reproduce the observations on lysimeters very well. One advantage of this approach is also the limited number of parameters, which is important for regional or global application of SVAT model.

In your detailed review, you suggested to test an intermediate approach that has a better physical consistency. We have tested the HCC functions of Iden et al., 2015s. This corrected form , which further clarifies our statement is unknown to hydrometorological and climate modelers. We are very pleased to be the first to include this formulation in a physical land surface model used in both hydrology and climate research. This approach gives very promising results.

So we propose to include this new approach (called VGc) in this detailed answer and in the revised version of our article. To this end, we modified the figures by adding a new experiment called VGc in addition to the 3 previous functions used.

We also proposed to modify the title of the article like this : "Evaluation of four Hydraulic Conductivity Curves in the Land Surface Model ISBA with data from several lysimeters."

For example, the 4 following figures show the results obtained with  VGc with hc=1cm, as suggested by Iden et al., 2015.
Figure R3.1 shows that the general agreement of VGc with the observation is good.
Figure R3.2 is the new version Figure 10 of the article. Figure R3.3 is the new version ofs Figure 11. The agreement of VGc with the soil saturation profile  is good. In June 2016 the soil profile is a little more saturated for VGc than for BCVG and drainage is similar for O1 lysimeter, and a little more reactive for G2 lysimeter. However, in terms of statistics, the two approaches are  similar, and a bit weaker than for BCVG during the event of February 2016.
Figure R3.4 shows the statistical results for all events and all variable, with  good results obtained by VGc.
We proposeto add this new approach in all the  figures of the revised version of the article.

In the VGc simulation, we used the  fitted n value of Van Genuchten with no restriction to n=1.1 as in the VG80 simulation. This might leads to differences in the matrix potential compared to VG.

The results  show that this approach is a real asset for this study.

[Figure]

Figure R3.1 : Chronicles of Water Mass, water volumetric content at 50cm, and Drainage for each lysimeters, for observation, BCVG and VGc.

[Figure]

Figure R3.2 : Daily precipitation (mm.day−1 ), hourly effective wetting saturation profile (%) observed (OBS) and simulated by BC66, VGc and BCVG, and daily drainage (mm.day−1 ) observed (in black), and simulated by BC66 in red, VG80 in blue, BCVG in green and VGc in orange during intense drainage in February 2016 for lysimeters O1 and G2. The Nash Sutcliffe efficiency (NSE) for each simulated drainage is also given.

[Figure]

Figure R3.3 : Daily precipitation (mm.day−1 ), hourly effective wetting saturation profile (%) observed (OBS) and simulated by BC66, VGc and BCVG, and daily drainage (mm.day−1 ) observed (in black), and simulated by BC66 in red, VG80 in blue, BCVG in green and VGc in orange during intense drainage in February 2016 for lysimeters O1 and G2. The Nash Sutcliffe efficiency (NSE) for each simulated drainage is also given.

[Figure]

Figure R3.4 : Taylor diagrams for hourly Total Water Mass (a), hourly Volumetric water content at 0.5m depth (b), Daily drainage (c) and Daily intense drainage events (d). Experiment BC66 is plotted in red, VG80 in blue, BCVG in green and VGc in orange. Additional experiments with an homogeneous soil profile (BCVGHOM) are represented as an open green circle, with parameters estimated from usual PTFs (BCVGP T F ) by a brown cross, and with parameters estimated with PTF except n estimated in situ (BCVGP T F n) by a purple cross. The Pearson correlation coefficient, the root-mean-square error (RMSE), and the normalized standard deviation are summarized in this diagram.

2. Equations:
The equations of the model used in the study must be either directly stated or clearly referenced. A statement such as "A complete description of the model equations used to simulate water transport can be found in xxxx (xxxx)" would be appropriate, especially for the soil hydrology processes that are the core of the paper. I have not found this. I understand that the mixed-form Richards equation was used, but with what kind of boundary conditions? This is especially of interest for the lower boundary condition, where Figs. 9, 10 and 12 indicate drainage under apparently unsaturated conditions, which is not possible when the lysimeters are operated with the traditional lysimeter boundary condition (i.e., seepage).

We agree with you. We 'll add the sentence proposed before each equation in the revised article version.

The boundary condition of the surface is the atmospheric boundary, and the boundary lower is free drainage.

Thanks for detecting an error in the plot: a 10-day lag between the water saturation profile and drainage in Figures 9,10, and 12 crept in for lysimeter G2. By correcting this error (see figures R3.2 & R3.3), the drainage appears well in phase with the saturation profile.

3. Data:

Which sensors were used? Since in science the reproducibility of results is of utmost importance, the high quality journals nowadays require that all necessary data are given. This bloats the manuscript itself too much, but it is possible to either put the data in data archives or publish them separately in appropriate data journals or attach them directly to articles as additional information. As a specific example, I would actually be interested in repeating the June 2016 G2 lysimeter scenario because the sudden breakthrough of water to the drainage seems difficult to explain with the VG80 data provided.

However, with the information currently available, it is not possible to perform such a test and judge the validity of the scientific statements.

On the GISFI site, TDR probes are RIME-PICO32 sensors with internal TDR-electronics. They are set horizontally and record the water content in cm3 cm− 3 ( ± 0.01) on an hourly basis. The calibration was performed on two measurements, one in dry and one in water-saturated condition. In the OPE site, soil moisture sensors used (UMP-1Umwelt Geräte Technik GmbH) are based on frequency domain reflectometry (FDR) method and measure local change in dielectric permittivity.

We will provide this information in the revised version of the article.

The data were provided by the GISFI and the OPE. The data can be accessible on requests, but, as the data are protected by a convention, the authors cannot provide access to the data. We will add contact coordinates so that anybody could ask access to these datasets.

4. Results.

Some of the results are shown as examples, which is fine. However, the reader should have access to the full results. So please use the supplementary material to list data not shown in the manuscript! For me, especially diagrams like figure 1 for lysimeters O1 and G2 are of great interest, also for the five other lysimeters.

We understand your point of view. We are committed to use the appendix or supplementary material to show the diagrams w-psi/w-k for each lysimeter and a table summarizing the parameters.

5. Lower boundary condition:
In line 181 authors state "By assuming the unit-gradient assumption, the drainage at 2 m can be considered equal to the hydraulic conductivity. ksat is then assumed equal to the observed drainage when the soil water content is saturated" --- The statement is correct if really unit-gradient conditions prevail in the lysimeter. In typical lysimeters, where the water flows freely into the atmosphere, this would require full saturation of the entire lysimeter(!). In suction controlled lysimeters it is different, but I did not find a statement about this in the paper. Do you have any indication that this assumption can be (approximately) true if the whole lysimeter is not saturated? Or any reference where the validity of this approximation has been shown?

Yes it is the case. In the Figure R3.5, variation of the total hydraulic pressure head between depths 50 and 150 cm ($\frac{dH}{dz}$) is represented for each lysimeter. Lines blue and red represent a gradient equal to 1 and 0 respectively. We can observe that the gradient is most often equal or very close to 1. Some variations exist, in particular for lysimeter G3, because of strong and deep roots. are presents. Nevertheless, when the saturation of volumetric content in the soil is reached, the gradient is equal to 1, and we can assume at these moments, ksat is equal to observed drainage.

[Figure]

Figure R3.5 : Variation of the total hydraulic pressure head between 50 and 150 cm ($\frac{dH}{dz}$) of depth is represented for each lysimeter. Lines blue and red represent a gradient equal to 1 and 0 respectively.

6. Conductivity functions.

Conductivity functions are key to the outcome of the simulations. As always with simulations, we can say, "What you put in determines what you get out." I would like to see the functions used in this comparison in appropriate graphs. The single Figure 1 is difficult to read and not sufficient. Please see my comments on Figure 1, below. Also, the coefficients of the hydraulic functions used in the simulation study should be listed in a table (perhaps in the supplemental material), as the information in Figure 2 is not sufficient to reproduce the functions.

We illustrated in Figure R3.6 the HCC estimated from volumetric water content at each depth and the drainage for at least 3 total years. As stated in the answer to your question #4, we'll provide all the plot in supplementary material of the revised article.

.

[Figure]

Figure R3.6 : Hydraulic conductivity curves for O1 lysimeters at different depths (20, 50,100 and 150 cm) . Observations are the points, curves fitted with BC66, VG80, VGc and VGBC are in red, blue, orange and green respectively. The black points are the mean of water content for each value of hydraulic conductivity

**COMMENTS RELALTED TO SPECIFIC TEXT PASSAGES**
**1. INTRODUCTION**
line 45. "Vereecken et al. (2019) suggested a number of directions for improvement: introduce more physical processes such [... ] improve the representation of [...] soil parameters," --- Agreed. But I do not really see the VGBC approach as an improvement of soil parameterizations in the above-mentioned sense.

You are right, VGBC is more an improvement of physical processes in ISBA compared to BC, since it helps better reproducing the water content close to saturation. This sentence will be modified, especially since we will introduce the VGc approach in the revised version of the article.

line 56 "These relationships are simple to parameterize and very stable numerically." --- There is some irony in the fact that Rien van Genuchten actually developed his parametrization to solve the problem of a discontinuous water capacity function that causes numerical problems in simulations with the

Richards equation. So, VG80 should be numerically stable except for cases with very small n.

Yes, we proposed to reformulate these sentences like : "However, the VG80 relationships are less stable than BC66 for small values of n."

line 60 "However, the VG80 relationships are less stable than BC66 for coarse-textured soil, mainly because of the complexity of the hydraulic conductivity function (Vogel et al., 2000)." --- In fact, I am not aware of this statement in the Vogel paper. I know VG80 is problematic for coarse grained soils in the dry moisture range, but in that range there is not much difference to BC66.

Yes, VG80 is problematic for coarse grained soils, with problems of stability. Even if the differences between the hydraulic conductivity of BC66 and VG80 are generally much less severe for coarse-texture soils.

line 73 "We derive their hydrodynamic parameters directly from observation" --- This is certainly the key to all subsequent results. As already indicated, the documentation on this important point is not sufficiently presented in the paper.

Thank you for these useful comments. Indeed, we did not provide details on the soil parameters calibration procedure, as we mainly focused on the difference between the soil parameters derived from in-situ data to those derived from pedotransfer functions. We will add the details in the revised version of the article.

The calibration of the soil parameters was performed using two methods: i) an objective least squares function which minimizes the sum of the squares of the deviations and corresponds to maximizing the likelihood with a normal distribution (function nls of rstudio). ii) the package SoilHyP (Dettmann et al., 2022) which uses the Shuffled Complex Evolution (SCE) optimization.

The two methods converge on very closed estimated parameters ( $\delta b: 1.6;\ \delta n: 0.01; \delta wsat: 0.0085\ (m^3 m^{-3}); \delta\alpha: 2\ (m^{-1}); \ \delta\Psi sat: 0.019\ (m)$ ).

Statistical scores and errors metrics on the estimation of the soil water retention curves (WRC) are presented in Table R3.1 for the BC66 and VG80 relationships at each depth. On the entire fit and close to saturation ($> \omega_{sat} * 0.9$), values median, minimal and maximal are presented. The fitted are generally better in depth, and a better r² for VG80 than for BC66 near to saturation. Near saturation, the NRMSE is also better for VG80 than BC66.

Table R3.1 : Statistical scores (regression, r²) and errors metrics (Normalized Root Mean Square Error, NRMSE, and Akaike information criterion, AIC) for

the BC66 and VG80 relationships at each depth (20-50-100 and 150 cm) on the total entire fit (Total) and close to saturation ($> \omega_{sat} * 0.9$) for all lysimeters for soil water retention curves (WRC).

**Total**

| | r² BC66 | r² VG80 | NRMSE BC66 | NRMSE VG80 | AIC BC66 | AIC VG80 |
|---|---|---|---|---|---|---|
| **20cm** | | | | | | |
| median | 0,891 | 0,886 | 0,337 | 0,34 | 1755,675 | 1777,015 |
| min | 0,846 | 0,84 | 0,229 | 0,189 | 23,154 | 34,243 |
| max | 0,958 | 0,966 | 0,393 | 0,401 | 5124,888 | 4938,271 |
| **50cm** | | | | | | |
| median | 0,69 | 0,84 | 0,67 | 0,4295 | 931,76 | 550,886 |
| min | 0,024 | 0,666 | 0,421 | 0,156 | -1045,18 | -1020,23 |
| max | 0,897 | 0,976 | 1,761 | 0,693 | 5828,86 | 5773,81 |
| **100cm** | | | | | | |
| median | 0,968 | 0,967 | 0,2285 | 0,1975 | -2003,996 | -2061,039 |
| min | 0,691 | 0,684 | 0,111 | 0,095 | -3454,685 | -3501,568 |
| max | 0,989 | 0,991 | 0,624 | 0,563 | 2368,327 | 2270,159 |
| **150cm** | | | | | | |
| median | 0,966 | 0,97 | 0,244 | 0,2435 | -1842,222 | -1546,381 |
| min | 0,711 | 0,66 | 0,122 | 0,159 | -5659,271 | -5651,963 |
| max | 0,991 | 0,987 | 0,608 | 0,7 | 6884,241 | 6852,803 |

**Close to Saturation**

| | r² BC66 | r² VG80 | NRMSE BC66 | NRMSE VG80 | AIC BC66 | AIC VG80 |
|---|---|---|---|---|---|---|
| **20cm** | | | | | | |
| median | 0,98 | 0,99 | 1,136 | 0,757 | -129,256 | -216,541 |
| min | 0,967 | 0,978 | 1,11 | 0,665 | -249,818 | -419,598 |
| max | 0,983 | 0,995 | 1,212 | 1,7 | 26,347 | 4,815 |
| **50cm** | | | | | | |
| median | 0,938 | 0,926 | 0,629 | 0,483 | 220,637 | -150,892 |
| min | 0,739 | 0,75 | 0,383 | 0,369 | -105,151 | -868,689 |
| max | 0,99 | 0,999 | 0,775 | 1,041 | 2154,151 | 1429,798 |
| **100cm** | | | | | | |
| median | 0,932 | 0,953 | 0,493 | 0,568 | 111,815 | -84,368 |
| min | 0,739 | 0,773 | 0,219 | 0,161 | -118,14 | -226,769 |
| max | 0,979 | 0,987 | 1,12 | 1,412 | 770,041 | 642,032 |
| **150cm** | | | | | | |
| median | 0,849 | 0,931 | 0,403 | 0,313 | 247,332 | 198,666 |
| min | 0,543 | 0,67 | 0,187 | 0,17 | -209,217 | -2331,309 |
| max | 0,964 | 0,996 | 0,598 | 1,02 | 720 | 696,064 |

The calibration of the hydraulic conductivity curves (HCC) was also realized. For each depth, we estimated the hydraulic conductivity at saturation with the water volumetric content and the drainage at 2m for at least 3 years  by averaging the drainage values for each water volumetric content value. Statistical scores are shown in Table R3.2 for the four relations. RMSE are better at depth with closer differences than at surface, and always better for VGBC (<0.5 mm/h). AIC are relatively closed between the four relations.

Table R3.2 : Statistical scores (regression, r²) and errors metrics (Root Mean Square Error, RMSE,  and Akaike information criterion,AIC) for the BC66 and VG80 relationships at each depth (20-50-100 and 150 cm) for calibration for hydraulic conductivity curve (HCC).

| | r² BC66 | r² VG80 | r² VGc | r² VGBC | RMSE BC66 | RMSE VG80 | RMSE VGc | RMSE VGBC | AIC BC66 | AIC VG80 | AIC VGc | AIC VGBC |
|---|---|---|---|---|---|---|---|---|---|---|---|---|
| **20cm** | | | | | | | | | | | | |
| median | **0,734** | 0,252 | 0,362 | 0,475 | 0,545 | 0,626 | 1,201 | **0,467** | -20,406 | **-230,292** | 92,441 | -35,782 |
| min | 0,318 | 0,214 | 0,153 | 0,242 | 0,476 | 0,546 | 0,604 | 0,427 | -34,891 | -272,773 | -28,618 | -67,644 |
| max | 0,766 | 0,361 | 0,415 | 0,482 | 0,928 | 1,008 | 5,049 | 0,767 | 39,966 | -24,493 | 178,235 | -32,912 |
| **50cm** | | | | | | | | | | | | |
| median | **0,601** | 0,295 | 0,415 | 0,332 | 0,669 | 0,504 | 0,758 | **0,426** | **-27,32** | -27,592 | -26,061 | -25,694 |
| min | 0,346 | 0,106 | 0,181 | 0,185 | 0,199 | 0,223 | 0,345 | 0,132 | -37,285 | -62,605 | -36,637 | -46,585 |
| max | 0,846 | 0,524 | 0,712 | 0,653 | 0,82 | 0,788 | 9,66 | 0,708 | 65,976 | -12,016 | 226,709 | -11,994 |
| **100cm** | | | | | | | | | | | | |
| median | **0,688** | 0,417 | 0,578 | 0,36 | 0,352 | 0,425 | 0,407 | **0,3** | -23,813 | -25,735 | -23,651 | **-30,998** |
| min | 0,398 | 0,125 | 0,208 | 0,19 | 0,283 | 0,366 | 0,352 | 0,187 | -63,182 | -255,385 | -152,687 | -65,784 |
| max | 0,808 | 0,876 | 0,876 | 0,888 | 0,649 | 0,659 | 0,809 | 2,034 | 51,406 | -16,898 | 65,95 | 142,02 |
| **150cm** | | | | | | | | | | | | |
| median | **0,599** | 0,248 | 0,472 | 0,307 | 0,548 | 0,659 | 0,612 | **0,485** | **-34,007** | -27,345 | -30,193 | -32,703 |
| min | 0,194 | 0,132 | 0,155 | 0,184 | 0,298 | 0,366 | 0,357 | 0,175 | -59,104 | -468,014 | -32,471 | -88,413 |
| max | 0,96 | 0,584 | 0,801 | 0,667 | 0,747 | 0,742 | 0,946 | 0,675 | 59,053 | -17,797 | 81,5 | -21,512 |

**2. EXPERIMENTAL PROTOCOL**

- line 85 "These two sites are separated by a distance of 97 km." --- Please state here the basic hydro-meteorological parameters: total precipitation (comes too late at the end of the section), total estimated ETp, height above sea level, mean temperature.

We propose to move the paragraph from line 110 after the first paragraph of this section by adding the information on height above seal level, mean temperature :

" The height above sea level is 350 and 224 m for OPE and GISFI sites, respectively. The sets of atmospheric forcing variables (wind speed, precipitation rate, short-wave incident radiation, air temperature, air humidity, atmospheric pressure) are observed in situ at an hourly time step by two local meteorological stations, one at OPE and one at GISFI. Long-wave radiation is derived from the equation of Prata (1996). Atmospheric data gaps are filled by regressions on available data using two neighboring meteorological stations. The gaps represent up to 12% of the observations for the GISFI site (Table 2). The mean temperature is 10.8 °C and 10.2 °C for the GISFI site.
Annual precipitation is 20 % higher at OPE compared to the GISFI experimental station (876 mm.year$_{-1}$, and 727 mm.year$_{-1}$, respectively, Table 2). The atmospheric forcing is assumed to be identical for all the lysimeters of each site."

- line 101 "with different layers of limestone more or less cracked" --- uuh, taking an undisturbed lysimeter in such material is difficult. Described somewhere in literature?

For the extraction of exact soil monoliths, the procedure used is the one developed by Umwelt-Geräte-Technik GmbH and proven nationally and internationally. The core can be extracted with high precision and without disturbing the soil structure without the use of heavy extraction technology.
Please, see this website where all the information are described :
https://www.ugt-online.de/en/products/lysimeter-technology/excavation-techniques

- line 104 "They are equipped with suction and temperature probes as well as time-domain reflectometry (TDR) probes" --- please specify the used instruments/sensors.

Each lysimeter was placed on weighting cells (10 g resolution). Tipping counters (Umwelt Geräte Technik GmbH) were used for drainage monitoring. Matrix potential was measured through suction in ceramic filters by tensiometers (Tensio 160, Umwelt Geräte Technik GmbH). On the GISFI site, TDR probes are

RIME-PICO32 sensors with internal TDR-electronics. They are set horizontally and record the water content in cm3 cm− 3 ( ± 0.01) on an hourly basis. The calibration was performed on two measurements, one in dry and one in water-saturated condition. In the OPE site, soil moisture sensors used (UMP-1Umwelt Geräte Technik GmbH) are based on frequency domain reflectometry (FDR) method and measure local change in dielectric permittivity.

We'll add these details in the full text.

- line 113 "The gaps represent up to 15% of the observations for the GISFI site (Table 2)." --- I tried to figure that out from table 2, but find there a different value, 12%.

Sorry for this mistake. The good value was provided by Table 2, the gap of the observation for the GISFI is well 12%. Thank you for this comment.

- Line 115 "atmospheric forcing" – how expressed? How is the upper soil hydraulic boundary condition expressed in the model?

Atmospheric forcing includes all the variables cited:To be clearer, we'll precised hourly 10m wind speed (m/s), 2m air temperature & humidity (K, Kg/kg) precipitation rate Kg/m2/s),  short-wave incident and long wave radiations (W/m2), surface atmospheric pressure (Pa) .

As ISBA is a SVAT scheme, the concept of upper soil hydraulic boundary conditions is not often considered like that. But, it can be stated that the upper soil hydraulic boundary conditions expressed in ISBA  are variable flux boundary conditions.

ISBA compute the water and energy balance at a 5-minute time step. The precipitation can first be intercepted by the vegetation. Then, the water that drops to the soil can infiltrate. The Green-Ampt approach is used to determine the maximum amount of water that infiltrate the soil. If the precipitation cannot infiltrate, there is some runoff. In the present study, as the maximum rate of precipitation is not that much, and the area is flat, there is no runoff.

- line 117: "ISBA model" --- I would like to see some more equations related to the main hydrological processes in the soil. For example, I have not found the implementation of the boundary conditions, and perhaps the calculation of isothermal vapor conductivity via a function of soil texture could somehow be shown without having to consult Braud et al. (1995).

The most important equation is Richards' equation, that's why we put it in front. We agree to add some new equations in the  appendix for more comprehension.

For the Implementation of the bottom boundary condition, we just stated that it's a free drainage condition.

The isothermal vapor conductivity is active when the soil is close to the wilting point, and therefore when the soil is dry, which is not our case; therefore we do not find it necessary to put the equation in this article. We propose to remove its mention in the article.

- line 154. As a soil hydrologist, I am somewhat surprised at this type of notation. Why psi(w) and K(psi), instead of expressing both w and K as a function of psi (which is common, at least in the soil hydrology community)? Of course, this is meant only as a comment and is not of any consequence....

This seems to depend on the scientific community. In the atmospheric community,  these relations are generally expressed in this way (Viterbo and Beljaars, 1995, Decharme et al., 2011).

- line 157: "the more complex closed-form equations from van Genuchten (1980)" --- No. The equations may look a bit more complicated, but VG's approach is certainly not "more complex". The fact that a simple algebraic equation is very short in one case, while it looks a bit more complicated in the other case, should not be interpreted as "complex" in 2022.

We agree and we will remove this expression.

- line 162 "α $(m_{-1})$ [is] the inflection point where the slope of the soil-water retention curve (dω/dψ) reaches its maximum value," --- This is wrong. $1/\alpha$ is in between the inflection point and the air-entry value.

Yes, you are right.

- "line 163. "l [is] the Mualem (1976) dimensionless parameter that determines the shape of the hydraulic conductivity curve" --- No, It does not determine the "shape" of the conductivity curve, but rather the slope of the log K vs. log psi curve in the unsaturated range. The "shape" of the function is unaffected, log K (log psi) remains linear.

Yes, you are right. We proposed to simply states that  Mualem (1976) dimensionless parameter that determines the slope of the hydraulic conductivity curve"

**3. ESTIMATION OF PARAMETERS**

- Line 173 "The rich data sets collected by the lysimeters allow the derivation of the soil hydrodynamic parameters"; "For instance, Fig. 1 plots" --- Figure 1 is central to the entire paper and its findings. I have a number of comments on it, which are included at the end of my comments here. These data must be given for all lysimeters, as they determine the outcome of the simulations and thus affect all conclusions of the paper.

Please, see above (response to point of line 73).

- line 179: "$\omega_{sat}$ is determined by the 99th percentile of the observed soil volumetric water content" --- to be sure: this is 6 years * 365 days * 24 values = > 52'000 water content data for each layer? Just make this clear.

Yes. That's correct. We propose to add the number of observations in the Table 2.

- line 178: "observed $\psi$-$\omega$ relationship at each depth" --- I assume the depths listed in Table 2? Are the layer boundaries of the different materials in the simulation chosen to be at the mean distance between these observation depths?

Yes. That's correct, the measurement depths are given in Table 2. The soil resolution in the model was missing, and we'll add it: 13 soil layers are used, with nodes at 0.01, 0.04, 0.1, 0.2, 0.4, 0.6, 1.0, 1.2, 1.4, 1.6,1.8 and 2 m.

- line 182 "By assuming the unit-gradient assumption, the drainage at 2 m can be considered equal to the hydraulic conductivity." --- Did you ever observe fully saturated unit gradient conditions in the lysimeter? Saturation just at the base is not a sufficient condition.

Please see above my answer to your question 5.

- line 193" Vogel et al. (2000) determined a limit of $n < 1.3$ below which Eq.(5) is numerically unstable." – Well, that's a complex issue. See remarks at the begin of the review.

Yes.That's correct. Please see the response in the part Specific Comment Basic Approaches.

- line 195. "The l parameter in Eq. (5) from VG80 is estimated with a simple calibration via ISBA sensitivity experiments with l ranging from -5 to 5. " --- This is incomprehensible: What exactly was done in this "simple calibration via ISBA sensitivity experiments"? Also, it is suspicious that the optimized values for the GISFI lysimeters are all at the limit of a permissible range. Furthermore, with l= -5, we are outside the physically permissible range for many sets of hydraulic

functions and might face even increasing conductivity in dry soil (see Peters et al., 2011).

Thank you for raising this point. Although a negative l is losing its physical meaning, such negative values are often used as notices by reviewers 1 and 2.. Therefore, we span all the values between -5 and 5 to calibrate this parameter. The NRMSE score on the comparison between observed and simulated drainage of the sensitivity tests are presented in Table R3.4. Best results were obtained for a tortuosity fixed at 0.5 for OPE lysimeters (O1-O2 and O4); and -5 for GISFI lysimeters (G1-G2-G3-G4).

Table R3.4 : Normalized Root Mean Square Error (NRMSE) scores from simulations with VG80 for each lysimeters, with variation of the parameter l.

| | NRMSE | | | | | | |
|------|-------|-------|-------|-------|-------|-------|-------|
| l | G1 | G2 | G3 | G4 | O1 | O2 | O4 |
| -5 | 0,738 | 0,851 | 0,861 | 0,81 | 0,867 | 0,929 | 0,777 |
| -2 | 0,741 | 0,875 | 0,989 | 0,813 | 0,861 | 0,946 | 0,753 |
| -1 | 0,763 | 0,93 | 0,968 | 0,81 | 0,873 | 0,931 | 0,763 |
| -0,5 | 0,738 | 0,898 | 0,969 | 0,855 | 0,873 | 0,926 | 0,763 |
| 0,5 | 0,774 | 1,038 | 0,876 | 0,817 | 0,694 | 0,817 | 0,708 |
| 1 | 0,929 | 1,185 | 0,899 | 0,826 | 0,812 | 0,826 | 0,715 |
| 2 | 0,878 | 1,144 | 0,861 | 0,849 | 0,694 | 0,824 | 0,708 |
| 5 | 0,966 | 1,031 | 0,948 | 0,869 | 0,77 | 0,83 | 0,736 |

- line 200: "The derived parameters are presented in Fig. 2" --- As mentioned above, the parameters ($\omega_{sat}$, $\psi_{sat}$, $\alpha$, n, b, l and $k_{sat}$) should be listed (additionally) in a Table, maybe in supplementary material.

Yes.The values are given in the Figure 2 of the article. We agree to add also the derived parameters in a Table in supplementary material.

- line 215 "if the volumetric water content presents a slow decrease in summer at a given depth, it is considered that the roots have not yet reached this depth. The root depth is thus fixed at 2 m for lysimeters G3 and O2" --- Hm, this is really a very large depth for the grass roots. Unusual.

The lysimeter G3 is covered by dense alfalfa (Mathers et al., 1975, Johnson et al., 1998). For O2 lysimeter, vegetation is not controlled and there was not only grass, but a mixed of vegetation. We imposed a maximal depth of root, where a root density profile is defined according to Jackson et al.,1996, with roots denser at the surface than at the maximal depth.

- line 216 "varies for lysimeters G4, O1 and O3." --- Would be nice to see this illustrated.

Please see  table R3.5 that provides the maximal root depth imposed for these lysimeters.
The roots in ISBA have a density profile: there are more root in the top soil, but, although there are less root at the maximum root depth, their impact can be seen in the soil moisture during the dry period.

| Table R3.5 : Evolution of the root depth (cm) | | | |
|---|---|---|---|
| Depth (in cm) | 2009-2013 | 2013-2016 | 2018-2020 |
| G4 | 1.5 | 0.4 | ■ |
| O1-O3 | ■ | 0.8 | 2.0 |

- line 222: "(not shown") – Why not shown? In digital format, we do not have space limits in the supplemental information!

It was not shown to keep the article short, but, it is true this can be put in the supplementary info. We propose to add the following figure with a mean annual cycle of the LAI simulated for lysimeters with vegetation (Figure R3.7) and also the daily simulation of the  simulated LAI (Figure R3.8).

.

[Figure]

Figure R3.7 : Mean LAI cycle simulated for lysimeters with vegetation for each lysimeter

[Figure]

Figure R3.8 : Daily evolution of the LAI simulated for each vegetated lysimeter

**4. RESULTS**

- line 240 "we arbitrarily consider that the initial observed and simulated total water masses are equal for the BC66 experiment" --- ... and how do you get the initial simulated total water mass? Based on what initial conditions in the simulation? And is the initial water content of the VG80 identical to that of the BC66?

A one-year simulation spin-up is performed (we'll add this information in the text) and the outputs of the simulation are used as initial conditions. The hypothesis we made, to consider that the initial total mass in BC66 is equal to the observed makes it possible to estimate a mass of the "dry lysimeter", ie, the lysimeter without any water: total mass=dry mass + total mass of water estimated by BC66. Then we used this dry mass to estimate the total mass of VG80 by considering the same dry mass, but accounting for the total water mass simulated by VG80.

- line 249 "BCVG experiment obtains better scores than the other experiments in more than 42 % of cases" --- Certainly true, but the differences between the model variants seem to be insignificant. With the exception of lysimeter G4, where the BC66 parameterization leads to a different picture than the others, we can say that the variants give more or less the same results. The fact that in G3 all three model variants overestimate the water mass might hint on a problem with the initial conditions or the soil mass.

Yes you are correct. This is why for the total mass, we added comments on square correlation $R^2$, since the correlation doesn't depend on the initial mass hypothesis. It can also be seen that although the initial value was derived from the BC66 experiment, the biases are similar for the 3 experiments. The important overestimation of simulated mass with lysimeter G3 can be explained by the fact that alfalfa was planted during the observed period, and it can generate an important biomass as well as important variation of mass between minimal and maximal values, and so the differences are more important.

- line 258 "VG80 obtains weaker statistical scores in 96 % of the cases, because soil water saturation is reached too rapidly"--- This could be an indication of an incorrect conductivity curve. As mentioned in the introductory section of this study, the use of VG80 is not acceptable for such small values of n, and this result is a strong indication of an incorrect

conductivity curve. The study should be supplemented with a modified VG80 conductivity as suggested above.

Yes, you are right, there is a clear improvement with the VGc simulation (see figure R3.1-2-3-4).

**Table 1**

Please add some more key information: USDA soil type (e.g., sandy loam), monolithic or filled, bulk densities.

We proposed to modify the Table R3.6 as follow : Description of lysimeters : filling method, soil type, vegetation cover, number of texture observations, and textures ( in % of clay sand, and silt) ,at different depths. HyCa stands for Hypereutrci Cambisol.

Table R3.6

**Table 1.** Description of lysimeters: filling method, soil type, vegetation cover, number of texture observations, and textures (in % of clay, sand ans silt) at different depths. HyCa stands for Hypereutric Cambisol

| Site | GISFI experimental station | | | | OPE experimental station | | |
|---|---|---|---|---|---|---|---|
| Lysimeters | G1 | G2 | G3 | G4 | O1 | O2 | O3 |
| | Fill | Fill | Fill | Monolith | Monolith | Monolith | Monolith |
| Soil | Technosol | Technosol | Technosol | Cambisol | HyCa | Cambisol | HyCa |
| Soil cover | bare soil | bare soil | Alfalfa | Grass | Grass | Grass | Grass |
| Layers | 1 | 1 | 1 | 4 | 6 | 1 | 6 |
| Bulk density (kg.m$^{-3}$) | 1300 | 1300 | 1300 | 1300 | 1700 | 1700 | 1700 |
| USDA soil type | Sandy Clay Loam | Sandy Clay Loam | Sandy Clay Loam | Clay Loam | Clay | Clay Loam | Silty Clay Loam |
| (Sand,Clay,Silt) (%,%,%) | | | | | | | |
| Homogeneous | (61.6, 14.3, 24.1) | (61.6, 14.3, 24.1) | (62.4, 15.2, 22.4) | (32.0, 25.0, 43.0) | (3.0, 36.0, 61.0) | (31.0, 41.0, 28.0) | (18.0, 47.0, 36.0) |
| 0.2m | "" | "" | "" | (20.0, 15.0, 75.0) | (11.0, 4.0, 85.0) | (50.4, 18.0, 31.6) | (24.0, 28.0, 48.0) |
| 0.5m | "" | "" | "" | (17.0, 26.0, 57.0) | (0.0, 67.0, 37.0) | (42.0, 27.0, 31.0) | (16.0, 53.0, 31.0) |
| 1m | "" | "" | "" | (34.0, 33.0, 33.0) | (0.0, 19.0, 81.0) | (22.0, 6.0, 72.0) | (16.0, 53.0, 31.0) |
| 1.5m | "" | "" | "" | (56.0, 24.0, 20.0) | (0.0, 19.0, 81.0) | (22.0, 6.0, 72.0) | (16.0, 53.0, 31.0) |

**Table 2**

Layout is puzzling: two columns for GISFI water content sensors, one column for the respective matric potential sensors.

We try to avoid putting too much data by not repeating the values. With your comments, we proposed to modify the table.

Table R3.7 : Description of the available observations for each lysimeter: observation period, mean Annual precipitation (Precip) and drainage

(Drain). For each type of data, the available depths are indicated (cm). Quality of measurements is given as percentage of missing data: meteo gap for the meteorological forcing, defect for the lysimeters measurements.

**Table 2.** Description of the available observations for each lysimeter: observation period, mean Annual precipitation (Precip) and drainage water (Drain). For each type of data, the available depths are indicated (cm). Quality of measurements is given as percentage of missing data: meteo gap for the meteorological forcing, defect for the lysimeters measurements.

| Site | GISFI experimental station | | | | OPE experimental station | | |
|---|---|---|---|---|---|---|---|
| Lysimeters | G1 | G2 | G3 | G4 | O1 | O2 | O3 |
| Period | 2011-2016 | 2009-2016 | | 2011-2016 | 2014-2019 | | |
| Precip (mm.year$^{-1}$) | 727 | | | | 876 | | |
| Drain (mm.year$^{-1}$) | 317 | 337 | 115 | 170 | 312 | 304 | 363 |
| Total Water mass | full column | | | | full column | | |
| Volumetric water content | 100-150 | 50-100-150 | | 50 | 20-50-100-150 | | |
| Matric potential | 100-150 | 50-100-150 | | 50 | 20-50-100 | | |
| Drainage | 200 | | | | 200 | | |
| Temperature | 50-100-150 | | | | 20-50-100-150 | | |
| **Quality of data** | | | | | | | |
| Meteo gap (%) | 12 | | | | 10 | | |
| Defect (%) | 16 | 8 | 23 | 0 | 0 | | |

Fig.1
a) Are these data from five years in-situ measurements (hourly resolution)? Why is there no hysteresis?

It is a measurement of hourly resolution, but not on the entire chronicles because measurements of matric pressure are not always available. This represents on average 3220 of observations by lysimeters and by depth. There is no hysteresis because we averaged the values by pressure to have a better solution with the objective function used for estimated hydraulic properties.

b) How are the depicted conductivity data derived? From the paper, I learned that Ksat is derived from a unit-gradient assumption, and relative K-function in the unsaturated range is predicted by the BC66 reps. VG80 model.

With parameters estimated via the relation $\psi - \omega$, we can plot the relation between volumetric water content and the drainage, and thus estimate Ksat when $\omega = \omega_{sat}$. Such a method was then compared to the quantile 99.9 of hourly observed drainage value. Both estimations are in agreement.

c) Why do conductivities decrease with decreasing suction (@1 m, soil G2, grey dots)

This could be due to measurement errors.

d) How are the RETC data for O1 @1.5 m derived? Such experimental data cannot be obtained this far into the unsaturated range! Moreover, a water content of 35% is impossible at a suction of 1E+5 m. Note that this suction condition I beyond oven dryness, i.e., $\omega$ = 0!

Yes you are right. There is water content at this suction. There was an error on the data processing. Please see the updated figure below ( Figure R3.9.)

[Figure]

Figure R3.9 : Relationships between volumetric water content ($\omega$) and logarithm of the absolute value of the soil matric potential ($\psi$) for lysimeters O1 and G2. Observations at 0.2, 0.5, 1 and 1.5 m depth are in dot (orange, aquamarine, grey and purple respectively), estimations are in red and blue for BC66 and VG80 experiments, respectively.

e) What's the meaning of the dotted lines (I assume psi_s and $\omega$_s, but name it in the legend or caption!)

Yes, you are right, it is well $\psi_{sat}$ and $\omega_{sat}$. We will explain it clearly in the caption.

f) Why do we find such strongly different slopes of RETC in a packed lysimeter (G2) with homogeneous)!) soil? What are the bulk densities?

It is perhaps due to a modification of the soil during the filling process of the lysimeters. The bulk density is 1300 and 1700 kg/$m^{-3}$ for GISFI and OPE lysimeters, respectively.

g) Where are the data for O2, and O3, and for G1, G3 and G4? supplemental material.

We well put the corresponding figures in supplemental material.

**Fig. 2**

a) Ksat – I am not able to decipher the unit. 10₆ m/s ??

Yes, the unit is $10^6 m/s$. We'll improve the axis in the Figure.

b) The n values are very low, close to 1, but in this figure it is impossible to distinguish values near one. Please list the values also in a table.

Yes. As we said previously, we add the values in appendices.

**Fig. 9**

a) Why do we observe outflow in a system that does not reach saturation at the bottom? Are these suction lysimeters?

Thank you for your detailed reading. Indeed, there was an error in the plot: a 10-day lag between the water saturation profile and drainage in Figure R3.10 crept in for lysimeter G2. By correcting this error, the drainage appears well when the profile is at saturation.

n.

[Figure]

Figure R3.10 : Daily precipitation (mm.day−1 ), hourly effective wetting saturation profile (%) observed (OBS) and simulated by BC66, VG80 and BCVG,

and daily drainage (mm.day−1 ) observed (in black), and simulated by BC66 in red, VG80 in blue, BCVG in green and VGc in orange during intense drainage in February 2016 for lysimeters O1 and G2. The Nash Sutcliffe efficiency (NSE) for each simulated drainage is also give

**TYPOS AND MINOR ISSUES**

- Terminology. For me it is puzzling to speak of "experiments" when actually different simulations are meant. The "experiment" to me is the physical experiment where a system is manipulated to observe a response. Perhaps "model variants" or "model approaches" is more appropriate?

We understand your point of view. We proposed to change the term "experiments" with "model approaches".

- line 83: This is an international journal with many readers not familiar with french terrestrial research units: please define GISFI and OPE upon first occurrence.

GISFI : "French Scientific Interest Group on Industrial Wastelands"

OPE : "Perennial Environmental Observatory"

- line 95: which bulk density?

1300 and 1700 kg/$m^{-3}$ for GISFI and OPE lysimeters, respectively.

- line 200: Typo "lystimeters".

Thank you for this note.

- line 203: replace "deep" by "depth".

Thank you for this note.

- line 243 „average biases of 18.1 kg (19.18 and 21 for BC66 and VG80)" --- Keep an eye on your digits: Better "18.1 kg (19.2 kg and 21.0 kg for BC66 and VG80"

Thank you for this note.

- line 277 "if" ??????????

We want to say : "If the year 2016 is excluded, as a large rainfall event occurred in May_June 2016"

- line 304 "dynammic"

Thank you for this note.

**Literature cited**

Brooks, R. H. and Corey, A. T. Properties of Porous Media Affecting Fluid Flow, Journal of the Irrigation and Drainage Division, 92, 61–88,455 https://doi.org/10.1061/JRCEA4.0000425, 1966

Durner W. Hydraulic conductivity estimation for soils with heterogeneous pore structure. Water Resour Res, 30, 211–23. doi:10.1029/93WR02676, 1994

Ippisch O., Vogel H-J., Bastian P. Validity limits for the van Genuchten–Mualem model and implications for parameter estimation and numerical simulation. Adv Water Resour, 29, 1780–9. doi:10.1016/j.advwatres.2005.12.011, 2006

Jackson, R. B., J. Canadell, J. R. Ehleringer, H. A. Mooney, O. E. Sala, and E. D. Schulze, 1996: A global analysis of root distributions for terrestrial biomes. Oecologia, 108, 389–411, https://doi.org/10.1007/BF00333714.

Johnson, L. D., Marquez-Ortiz, J. J., Lamb, J. F. S., & Barnes, D. K. (1998). Root morphology of alfalfa plant introductions and cultivars. *Crop Science*, *38*(2), 497-502.

Kosugi K. Three-parameter lognormal distribution model for soil water retention. Water Resour Res, 30, 891–901, 1994

Malama B., Kuhlman KL. Unsaturated Hydraulic Conductivity Models Based on Truncated Lognormal Pore-Size Distributions. Ground Water, 53, 498–502. doi:10.1111/gwat.12220, 2015

Mathers, A. C., Stewart, B. A., & Blair, B. (1975). *Nitrate-nitrogen removal from soil profiles by alfalfa* (Vol. 4, No. 3, pp. 403-405). American Society of Agronomy, Crop Science Society of America, and Soil Science Society of America.

Peters, A., Durner, W., Wessolek, G. Consistent parameter constraints for soil hydraulic functions. Advances in Water Resources, 34(10), 1352-1365, 2011

Schaap, M. G., Van Genuchten, M.T. (2006). A modified Mualem–van Genuchten formulation for improved description of the hydraulic conductivity near saturation. Vadose Zone Journal, 5(1), 27-34.

van Genuchten M.T. A closed-form equation for predicting the hydraulic conductivity of unsaturated soils. Soil Sci Soc Am J, 44, 892–8, 1980

Vogel T., Cislerova M. On the reliability of unsaturated hydraulic conductivity calculated from the moisture retention curve. Transp Porous Media, 31, 15, doi:10.1007/BF00222683, 1988

Vogel T., van Genuchten M.T., Cislerova M. Effect of the shape of the soil hydraulic functions near saturation on variably-saturated flow predictions. Adv Water Resour, 24, 133–144. doi:10.1016/S0309-1708(00)00037-3, 2001

Viterbo, P., and A. C. M. Beljaars (1995), An improved land surface parametrization scheme in the ECMWF model and its validation, J. Clim., 8, 2716–2748, doi:10.1175/1520-0442(1995)008<2716: AILSPS>2.0.CO;2